# MITOL-dependent ubiquitylation negatively regulates the entry of PolγA into mitochondria

**Mansoor Hussain**, **Aftab Mohammed**, **Shabnam Saifi**, **Aamir Khan**, **Ekjot Kaur**, **Swati Priya**, **Himanshi Agarwal**, **Sagar Sengupta***

National Institute of Immunology, Aruna Asaf Ali Marg, New Delhi, India

☯ These authors contributed equally to this work.
* sagar@nii.ac.in

**Data Availability Statement:** All relevant data are within the paper and its Supporting Information files.

## Abstract

Mutations in mitochondrial replicative polymerase PolγA lead to progressive external ophthalmoplegia (PEO). While PolγA is the known central player in mitochondrial DNA (mtDNA) replication, it is unknown whether a regulatory process exists on the mitochondrial outer membrane which controlled its entry into the mitochondria. We now demonstrate that PolγA is ubiquitylated by mitochondrial E3 ligase, MITOL (or MARCH5, RNF153). Ubiquitylation in wild-type (WT) PolγA occurs at Lysine 1060 residue via K6 linkage. Ubiquitylation of PolγA negatively regulates its binding to Tom20 and thereby its mitochondrial entry. While screening different PEO patients for mitochondrial entry, we found that a subset of the PolγA mutants is hyperubiquitylated by MITOL and interact less with Tom20. These PolγA variants cannot enter into mitochondria, instead becomes enriched in the insoluble fraction and undergo enhanced degradation. Hence, mtDNA replication, as observed via BrdU incorporation into the mtDNA, was compromised in these PEO mutants. However, by manipulating their ubiquitylation status by 2 independent techniques, these PEO mutants were reactivated, which allowed the incorporation of BrdU into mtDNA. Thus, regulated entry of non-ubiquitylated PolγA may have beneficial consequences for certain PEO patients.

## Introduction

The central factor mediating mitochondrial DNA (mtDNA) replication is the sole catalytic subunit of mtDNA polymerase γ (PolγA). Together with the 2 identical subunits of the processivity factor (PolγB), PolγA forms the functional mtDNA polymerase (PolγA/B2) [1]. Mutations in *POLG*, the gene that codes for PolγA, are associated with mitochondrial disorders like progressive external ophthalmoplegia (PEO), Alpers–Huttenlocher syndrome (AHS), myocerebrohepatopathy spectrum (MCHS) disorders, myoclonic epilepsy myopathy sensory ataxia (MEMSA), and ataxia neuropathy spectrum (ANS). While the mutations in *PolG* are spread over the entire gene body, their frequency is more in the polymerase domain. Most of the common PolγA mutations (like A467T, W748S, and Y955C) have deficiency in catalytic binding and/or polymerase activity as demonstrated by *in vitro* assays [2–4]. It is assumed that mutant

**Funding:** Sagar Sengupta acknowledges National Institute of Immunology (NII) intramural funding and the following extramural funding sources: Department of Biotechnology (DBT), India (BT/MED/30/SP11263/2015, BT/PR23545/BRB/10/1593/2017, BT/PR27681/GET/119/269/2018), Council of Scientific and Industrial Research (CSIR), India (37(1699)/17/EMR-11), Science & Engineering Research Board (SERB), India (EMR/2017/000541) and J C Bose Fellowship (JCB/2018/000013). EK acknowledges DST Inspire Faculty Fellowship (DST/INSPIRE/04/2017/000088) for salary and funding. Shabnam Saifi acknowledges DBT-RA Fellowship (Batch 35/July 2019/10) for salary and funding. The funders had no role in study design, data collection and analysis, decision to publish, or preparation of the manuscript.

**Competing interests:** The authors have declared that no competing interests exist.

**Abbreviations:** AHS, Alpers–Huttenlocher syndrome; ANS, ataxia neuropathy spectrum; CHX, cycloheximide; DTT, dithiothreitol; DUB, deubiquitylating enzyme; IMM, inter-mitochondrial membrane; IPTG, Isopropyl-1-thio-β-D-galactopyranoside; LC1, light chain 1; MAVS, mitochondrial antiviral signaling; MCHS, myocerebrohepatopathy spectrum; MEMSA, myoclonic epilepsy myopathy sensory ataxia; MLS, mitochondrial localization signal; MPP, mitochondrial processing peptide; mtDNA, mitochondrial DNA; NHF, normal human fibroblast; OMM, outer mitochondrial membrane; PEO, progressive external ophthalmoplegia; PMSF, phenylmethylsulfonyl fluoride; PolγA, polymerase γ subunit A; qPCR, quantitative polymerase chain reaction; RT-qPCR, reverse transcription quantitative polymerase chain reaction; TCA, trichloroacetic acid; TIM23, translocase of the inner membrane 23; TOM, translocase of the outer membrane; UPS, ubiquitin proteasomal system; WT, wild-type.

PolγA patient variants also localize into the mitochondrial matrix as the mitochondrial localization signal (MLS) of PolγA is in the extreme N-terminus of the protein [5]. Hence, it is accepted that reductions in the PolγA activity within the mitochondrial matrix cause the dysfunctions associated with the abovementioned mitochondrial disorders. However, it has been reported that low levels of PolγA can also cause inefficient initiation of mitochondrial replication leading to phenotypes associated with PEO [6].

A vast majority of the 1,500 odd proteins which enter mitochondria use the pre-sequences of variable lengths which serve as the targeting motif. These proteins bind sequentially to the translocase of the outer membrane (TOM) complex situated in the outer mitochondrial membrane (OMM) and then to the translocase of the inner membrane 23 (TIM23) complex. Subsequently, the mitochondrial processing peptide (MPP) proteolytically removes the pre-sequences and releases the proteins into the mitochondrial matrix [7]. The intricacy of the regulatory process indicates the mitochondrial outer membrane as a vital regulatory hub for mitochondrial protein import and their subsequent functions.

Ubiquitylation is known to play important roles in multiple functions associated with mitochondria, namely mitochondrial homeostasis, mitochondrial mass, communication of mitochondria with other intracellular organelle, mitochondrial quality control, and mitophagy [8–10]. A variety of proteins involved in different mitochondrial biological processes as diverse as oxidative phosphorylation, Krebs cycle, and mitochondrial dynamics are ubiquitylated [11]. Inhibition of proteasomal degradation results in increased levels of mitochondrial proteins and abnormalities in mitochondrial morphology, thereby indicating to the possibility of uncontrolled protein import leading to the breakdown of the mitochondrial quality control system [12–14]. Hence, it has been hypothesized that when mitochondrial proteins are ubiquitylated, they are mistargeted, thereby compromising the quality of proteins entering the mitochondria [10]. Independent lines of evidences further link the ubiquitin proteasomal system (UPS) and mitochondria: (a) the recruitment of proteasome machinery to the surface of the mitochondria [15,16]; (b) the presence of several components of the ubiquitylation machinery, namely the E3 ligases (MITOL/MARCH5/RNF153; MULAN/MAPL; and PARKIN, RNF185, and KEAP1) and deubiquitylating enzymes or DUBs (USP30 in humans) on the outer membrane of the mitochondria [8]; and (c) the ubiquitylation inhibiting protein import into mitochondria in the yeasts [17].

MITOL/MARCH5/RNF153 (hereafter referred to as MITOL) with 4 membrane spanning segments belongs to RING-CH E3 ligase family. MITOL ubiquitylates, controls the levels, and consequently regulates the functions of a variety of substrates including mitochondrial fission factors like Drp1, Fis1, and MiD49; fusion proteins like Mfn1, Mfn2, and SLC25A46 [10]; and factors which induce mitophagy [18] and neuronal cell death and survival regulatory element, LC1 [19]. MITOL maintains the optimal mitochondrial quality control not only by specifically targeting mutant SOD1 and polyglutamine expanded protein [20,21] but also its own mutant counterpart [22]. Interestingly, MITOL regulates immunopathology during viral infection by targeting mitochondrial antiviral signaling (MAVS) protein and potentiating its degradation [23].

Here, we report that MITOL-mediated ubiquitylation negatively regulated the entry and functioning of PolγA inside the mitochondria. We determined the site on PolγA which was ubiquitylated by MITOL via K6 linkage. Ubiquitylated PolγA could not enter the mitochondria due to its decreased interaction with Tom20. A subset of PolγA mutant proteins expressed in PEO patients was prevented from entering into the mitochondria due to enhanced MITOL-dependent ubiquitylation. Instead, these PolγA variants became enriched in the insoluble fraction and demonstrated enhanced degradation. However, these PEO mutants could be reactivated by inhibiting their ubiquitylation, either by depleting MITOL or mutating the site on

PolγA (K1060) which gets ubiquitylated by MITOL. This allowed the PolγA mutants to enter mitochondria and thereby attain *in vivo* mtDNA replication levels near to that observed for wild-type (WT) PolγA. Overall, our results indicated that MITOL-dependent ubiquitylation act as a negative regulatory mechanism which fine-tuned the entry of PolγA into the mitochondria and thereby controlled its functions.

## Results

### PolγA levels are decreased by MITOL

Toward an effort to determine the mechanism of turnover PolγA, overexpression of 5 known mitochondrial E3 ligases (MITOL, PARKIN, MULAN, RNF185, and KEAP1) was carried out, and the levels of endogenous PolγA were determined. Only MITOL decreased the levels of PolγA (Fig 1A). MITOL overexpression also decreased the levels of TFAM, but not other proteins involved in mtDNA replication, namely PolγB or Twinkle (Fig 1B). The decrease in the level of PolγA was reversed upon MG132 treatment (Fig 1C) and occurred only in presence of wild-type MITOL (MITOL WT) and not its catalytically dead counterpart (MITOL CD) (Fig 1D). Depletion of MITOL, either transiently by 2 different MITOL siRNAs (Fig 1E) or by the stable expression of shMITOL in HeLa cells [24] (Fig 1F) or in 4-OHT treated MITOL[flox/flox] MEFs [24] (Fig 1G), all led to increased levels of PolγA and TFAM. However, the levels of other mitochondrial proteins (like Twinkle and Tom20) remained unaffected in absence of MITOL (Fig 1 E–G). The half-life of endogenous PolγA was increased upon depletion of MITOL as revealed by cycloheximide (CHX) chase experiment (Fig 1H and 1I). Importantly, neither the overexpression nor the ablation of MITOL had any effects on PolγA transcript levels (S1A and S1B Fig).

Next, we wanted to determine the binding parameters between PolγA and MITOL. Immunoprecipitation experiments indicated that MITOL interacted with PolγA (Fig 1J). Endogenous PolγA colocalized with endogenous MITOL in hTERT-immortalized normal human fibroblasts (NHFs) (S1C Fig). Similar colocalization was also observed for overexpressed Flag-tagged PolγA with Myc-tagged MITOL (S1D Fig). The spacer and thumb domains in PolγA encompassing amino acids 440–815 [25] (S1E Fig) interacted with MITOL. Reciprocally, the extreme carboxyl terminus cytosol-facing loop in MITOL (amino acids 253–278) interacted with PolγA (S1F Fig).

### MITOL ubiquitylates PolγA at K1060 via K6 linkage

Since the level of PolγA is regulated by MITOL, we next wanted to determine whether PolγA is ubiquitylated by this E3 ligase. *In vitro* ubiquitylation indicated that MITOL WT but not MITOL CD ubiquitylated PolγA (Fig 2A). This effect was also recapitulated *in vivo* using overexpressed His-tagged Ub, Flag-tagged PolγA, and Myc-tagged MITOL WT or MITOL CD (Fig 2B). Consequently, the shutdown of MITOL by using its cognate siRNA led to a decrease in the level of ubiquitylation of endogenous PolγA (Fig 2C). Next, we wanted to know the linkage via which MITOL ubiquitylated PolγA. It has been reported that MITOL ubiquitylates its substrates via K48 or K63 linkage [20,23,26]. However, analysis with WT "R" and "O" ubiquitin mutants (described in Materials and methods) indicated that PolγA were ubiquitylated *in vitro* by MITOL via K6 linkage (Fig 2D). We came to this conclusion because except K6R, for none of the other "R" mutants of PolγA, there is a loss of *in vitro* ubiquitylation of PolγA. This indicated that loss of ubiquitylation on PolγA is linked to the lack of lysine residue at K6 position of ubiquitin. Subsequently, when we used the K6O mutant (where only the lysine at sixth position on ubiquitin is present and all other lysines are mutated to alanines) in the *in vitro* ubiquitylation assay, there is a robust ubiquitylation of PolγA. It is to be noted that usage of

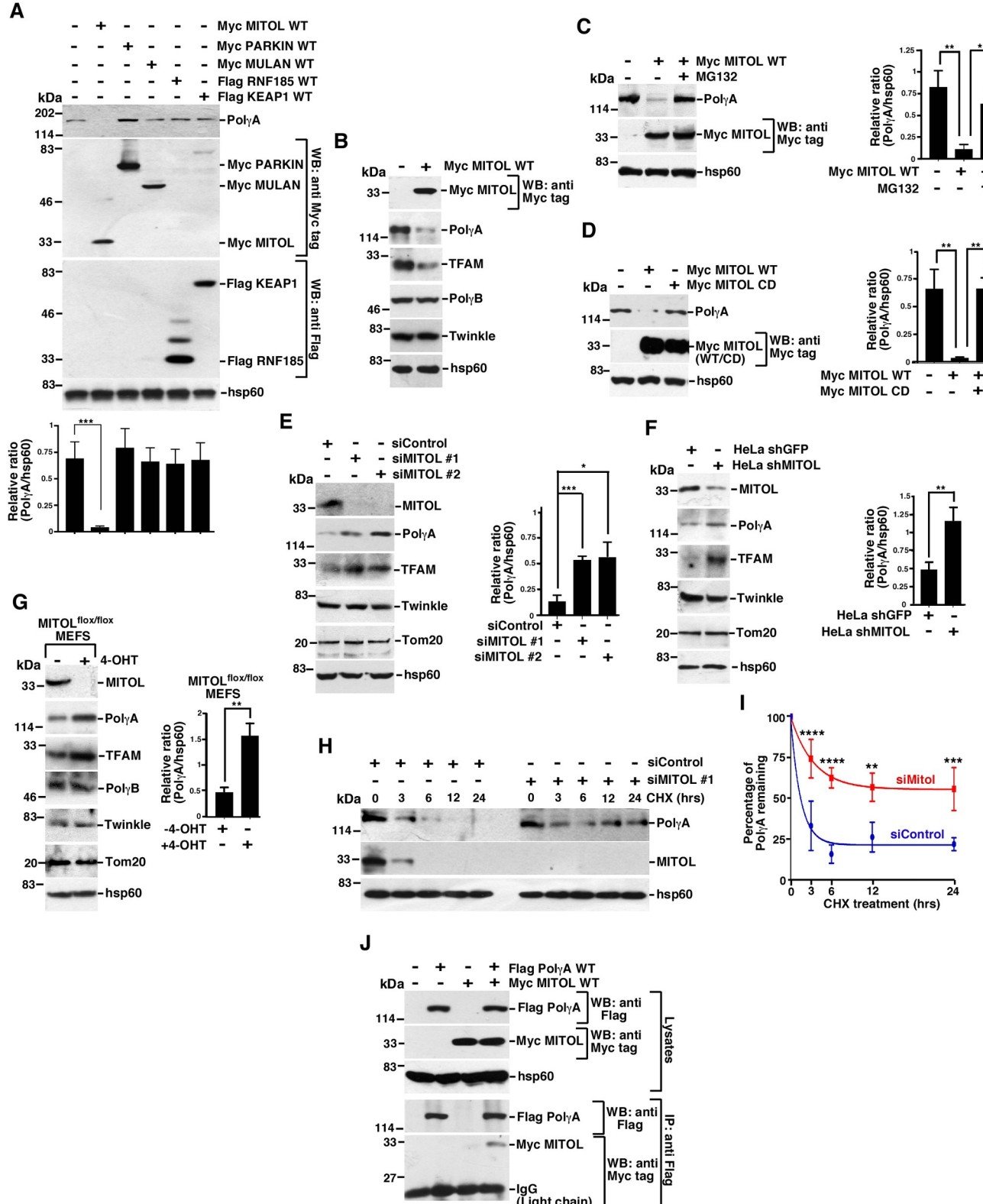

**Fig 1. MITOL interacts with PolγA. (A, B)** MITOL overexpression leads to decreased levels of PolγA. Whole cell extracts were made from HEK293T overexpressing the indicated mitochondrial E3 ligases. Western blot analysis was carried out with the indicated antibodies. The relative levels of PolγA to hsp60 in (A) have been quantitated from 3 biological replicates. **(C)** MITOL causes proteasomal degradation of PolγA. Whole cell extracts were made

from HEK293T cells overexpressing Myc MITOL for 24 hours and grown either in absence or presence of MG132. Western blot analysis was carried out with the indicated antibodies. The relative levels of PolγA to hsp60 have been quantitated from 3 biological replicates. **(D)** Catalytic activity of MITOL is essential for the proteasomal degradation of PolγA. Whole cell extracts were made from HEK293T cells overexpressing either Myc MITOL WT or CD. Western blot analysis was carried out with the indicated antibodies. The relative levels of PolγA to hsp60 have been quantitated from 3 biological replicates. **(E–G)** Depletion of MITOL stabilizes PolγA. Whole cell extracts were made from (E) HEK 293T cells transfected with either siControl or siMITOL #1 or siMITOL #2 (F) HeLa shGFP and Hela shMITOL cells or from (G) MITOL^flox/flox cells either untreated or treated with 4-OHT for 4 days. Western blot analysis was carried out with the indicated antibodies. For all 3 experiments, the relative levels of PolγA to hsp60 have been quantitated from 3 biological replicates. **(H, I)** Half-life of PolγA increases in absence of MITOL. (H) HEK293T cells transfected with either siControl or siMITOL #1. After 24 hours of transfection, cells were grown either without CHX treatment or after CHX treatment for the indicated hours (hrs), and whole cell extracts were made. Western blot analysis was carried out with the indicated antibodies. (I) The percentage of PolγA remaining after CHX treatment has been quantitated. The quantification is from 3 biological replicates. **(J)** PolγA interact with MITOL *in vivo*. (Top) Whole cell extracts were made from HEK293T cells transfected with Flag PolγA and Myc MITOL. Western blot analysis was carried out with the indicated antibodies. (Bottom) Immunoprecipitations were carried out with anti-Flag antibody using the respective lysates. The immunoprecipitates were probed with the indicated antibodies. Three independent biological replicates were carried out, and the same result was obtained. Numerical values for all graphs can be found in S1 Data. See also S1 Fig. CD, catalytically dead; CHX, cycloheximide; IgG, immunoglobulin G; PolγA, polymerase γ subunit A; WT, wild-type.

K48O or K63O ubiquitin mutants did not lead to PolγA ubiquitylation, as in these 2 ubiquitin mutants, the lysine at the sixth position in ubiquitin has been mutated to alanine.

Recently, K6- and K33 linkage–specific "affimer" reagents as high-affinity ubiquitin interactors have been characterized [27]. Using K6 linkage–specific affimer, we confirmed that K6 linkage on PolγA also occurred *in vivo* (Fig 2E). Further, we wanted to determine the specificity of MITOL-dependent ubiquitylation on PolγA. We found that the absence of HUWE1, an E3 ligase known to ubiquitylate its substrates via K6 linkage [27], could not increase the level of endogenous PolγA (S2A Fig). This indicated that HUWE1 did not ubiquitylate and thereby degrade PolγA. It has been recently reported that MITOL ubiquitylates multiple mitochondrial proteins, thereby preventing their import [28], while the deubiquitinase USP30 deubiquitylates these substrates, thereby promoting their entry [28,29]. We wanted to test whether USP30 could deubiquitylate MITOL-mediated ubiquitylation on PolγA. Recombinant USP30 (S2B Fig) was added either during (called simultaneous) or after (called sequential) MITOL-mediated *in vitro* ubiquitylation assay. In both the conditions, USP30 could deubiquitylate PolγA (S2C and S2D Fig). However, in asynchronously growing cells, co-expression of USP30 with MITOL could not revert MITOL-mediated decrease in PolγA levels (S2E Fig) nor revert the extent of ubiquitylation on PolγA (S2F Fig). This indicates that there are yet unknown factors or growth conditions which possibly regulate the ubiquitylation–deubiquitylation cycle of PolγA inside the cells.

Next, we wanted to determine the site(s) on PolγA which were ubiquitylated by MITOL. Using a combination of 2 independent ubiquitylation site prediction algorithms (UbPred and UbiPred) and a database (PhosphoSitePlus), a number of lysines were predicted on PolγA which could potentially be ubiquitylated by MITOL. UbPred, UbiPred, and PhosphoSitePlus predicted that PolγA may be potentially ubiquitylated by MITOL on 3 lysine residues. *In vitro* ubiquitylation with lysine to alanine mutants of PolγA indicated that only one of the residues (K1060) was ubiquitylated by MITOL *in vitro* (Fig 2F). This was validated *in vivo* when PolγA K1060R mutant showed complete loss of MITOL-mediated ubiquitylation (Fig 2G). Interestingly, PolγA K1060R interacted with MITOL as well as WT PolγA (Fig 2G), thereby indicating that MITOL could still bind with its substrates in which the site of ubiquitylation had been mutated. This phenomenon, maybe due to an attempted compensatory mechanism of MITOL on PolγA, has also been earlier observed for another MITOL substrate [26].

## Ubiquitylated PolγA cannot enter mitochondrial matrix

Having determined that PolγA was ubiquitylated by MITOL, we took the next logical step to understand whether this ubiquitylation affected its entry into the mitochondria. We first used

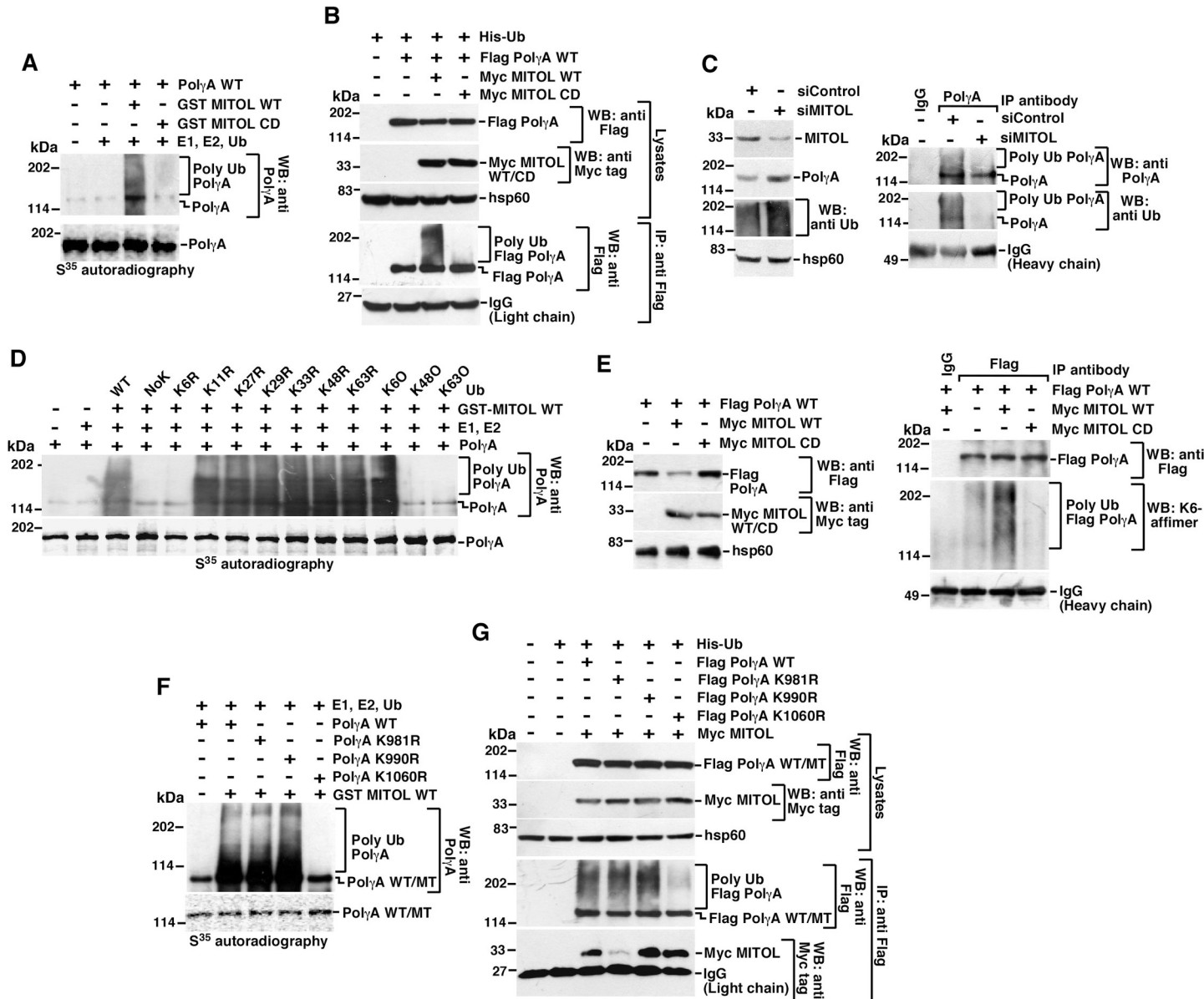

**Fig 2. MITOL ubiquitylates PolγA at specific residues via K6 linkage. (A)** MITOL ubiquitylates PolγA *in vitro*. *In vitro* ubiquitylation reactions were carried out using PolγA as the substrate and MITOL WT or CD. Post-reaction, the products were detected by western blot analysis with the indicated antibodies. Three biological replicates were carried out, and the same result was obtained. **(B)** MITOL ubiquitylates PolγA *in vivo*. (Top) Whole cell extracts were made from HEK293T cells transfected with His-Ub, Myc MITOL WT, or CD and Flag PolγA. Western blot analysis was carried out with the indicated antibodies. (Bottom) Immunoprecipitations were carried out with anti-Flag antibody, and the immunoprecipitates were probed with the indicated antibodies. Three independent biological replicates were carried out, and the same result was obtained. **(C)** Lack of MITOL prevents PolγA ubiquitylation. (Left) Whole cell extracts were made from HEK293T cells transfected with siControl or siMITOL. Western blot analysis was carried out with the indicated antibodies. (Right) Immunoprecipitations were carried out with anti-PolγA antibody, and the immunoprecipitates were probed with the indicated antibodies. Three independent biological replicates were carried out, and the same result was obtained. **(D)** MITOL ubiquitylates PolγA *in vitro* via K6 linkage. Same as (A), except the indicated "R" or "O" mutants of ubiquitin, were used during the reactions. Three independent biological replicates were carried out, and the same result was obtained. In "R" ubiquitin mutants, only a particular lysine residue in ubiquitin is mutated, while in "O" ubiquitin mutants, only 1 lysine is present on ubiquitin, and all other 6 lysines are mutated. **(E)** MITOL ubiquitylates PolγA *in vivo* via K6 linkage. Whole cell extracts were made from HEK293T cells transfected with MITOL WT or CD in presence of Flag PolγA. (Left) Lysates made were probed with the indicated antibodies. (Right) Immunoprecipitations were carried out with anti-Flag antibody (or the corresponding IgG), and the immunoprecipitates were probed with K6 affimer and the indicated antibodies. Three independent biological replicates were carried out, and the same result was obtained. **(F)** MITOL ubiquitylates PolγA *in vitro* at specific residues. *In vitro* ubiquitylation reactions were carried out using PolγA WT and 3 PolγA mutants, namely PolγA K981R, PolγA K990R, and PolγA K1060R as the substrates. Post-reaction, the products were detected by western blot analysis with anti-PolγA antibody. Three independent biological replicates were carried out, and the same result was obtained. **(G)** MITOL ubiquitylates PolγA *in vivo* at specific residues. (Top) Whole cell extracts were prepared form HEK293T transfected with His-Ub, Myc MITOL, and Flag PolγA WT or PolγA K981R or PolγA K990R, or PolγA K1060R. Western blot analysis were carried out with the indicated antibodies. (Bottom)

Immunoprecipitations were carried out with anti-Flag antibody, and the immunoprecipitates were probed with the indicated antibodies. Three independent biological replicates were carried out, and the same result was obtained. See also S2 Fig. CD, catalytically dead; His-Ub, His-tagged ubiquitin; IgG, immunoglobulin G; PolγA, polymerase γ subunit A; WT, wild-type.

both the Flag-tagged and untagged ubiquitylated PolγA and showed that both the versions have equivalent levels of entry into the mitochondrial matrix (Fig 3A and 3B). However, compared to the ubiquitylated variant, non-ubiquitylated PolγA entered the matrix with much better efficiency (Fig 3C and 3D). It was noted that the entry of PolγA did not occur in presence of either CCCP (an uncoupler which dissipated the mitochondrial potential) or Triton X-100 (which solubilized the inner membrane), thereby indicating the authenticity of the entry of PolγA into mitochondrial matrix. To investigate in more detail how ubiquitylation regulated the entry of PolγA, we next carried out an import assay with either PolγA WT or PolγA K1060A (which cannot be ubiquitylated by MITOL, S3A Fig). PolγA K1060R entered mitochondrial matrix with much greater efficiency than PolγA WT (Fig 3E and 3F).

To investigate in more detail how ubiquitylation regulated the entry of PolγA into mitochondria, we carried out ubiquitylation of PolγA for either 5 or 20 minutes (S3B Fig). Mitochondrial import assays for PolγA were carried out with the above reaction products. PolγA which has been ubiquitylated for 5 minutes could enter inside the mitochondria (i.e., the post-trypsin fraction) with much better efficiency (Fig 3G). This indicated a reciprocal relationship between MITOL-dependent ubiquitylation and the mitochondrial entry of PolγA.

Next, we wanted to determine how PolγA ubiquitylation by MITOL affected its entry into the mitochondria. We hypothesized that ubiquitylation of PolγA may negatively affect its binding to Tom20. Ubiquitylated or non-ubiquitylated PolγA were generated by in vitro ubiquitylation reactions carried out with either MITOL WT or MITOL CD (S3C Fig). Using these products, in vitro interaction experiments were carried out with either GST or GST-Tom20 (S3D Fig). It was revealed that non-ubiquitylated PolγA interacted better with Tom20 (Fig 3H). To further probe whether K6-linked ubiquitylation of PolγA was specifically involved, in vitro interaction with GST-Tom20 was carried out with PolγA ubiquitylated by MITOL using either K6R ubiquitin (which does not allow PolγA ubiquitylation, see Fig 2D) or K6O ubiquitin (which allows PolγA ubiquitylation, see Fig 2D). PolγA interacted better with Tom20 when it was generated by using K6R ubiquitin and not when using K6O ubiquitin (Fig 3I, input control in S3E Fig). This indicated that non-ubiquitylated PolγA binds better with Tom20, thereby allowing its better import into the mitochondrial matrix.

## PEO mutants have compromised mitochondrial entry and functions

To understand whether the enhanced entry of the non-ubiquitylated PolγA into mitochondrial matrix have any pathological implication, 4 PEO patient mutations were chosen, all of which led to their respective missense mutations (S1 Table). Using in vitro transcribed and translated PolγA WT and 4 PEO missense mutants (S4A Fig), it was determined by mitochondrial import assay that PEO mutants #1 and #2 entered mitochondrial matrix with drastically lesser efficiency compared to PolγA WT and the PEO #3 and PEO #4 patient mutants (Fig 4A and 4B, S4B Fig). Using 2.1–2.4-fold overexpressed PolγA WT and patient mutants (S4C Fig), the 2 PEO mutants which could not enter the mitochondrial matrix (i.e., #1 and #2) were found to be ubiquitylated to a higher extent compared to PolγA WT (Fig 4C, bottom panel). In contrast, 2 other PEO mutants (i.e., #3 and #4) were ubiquitylated to a similar extent as PolγA WT (Fig 4C, bottom panel; S4D Fig, right panel). Parallel in vitro ubiquitylation reactions with Ub WT and K6O ubiquitin indicated that like PolγA WT, PEO mutants #1 and #2 were also ubiquitylated via K6 linkage (Fig 4D). Consequently, at early time points (see Materials and methods

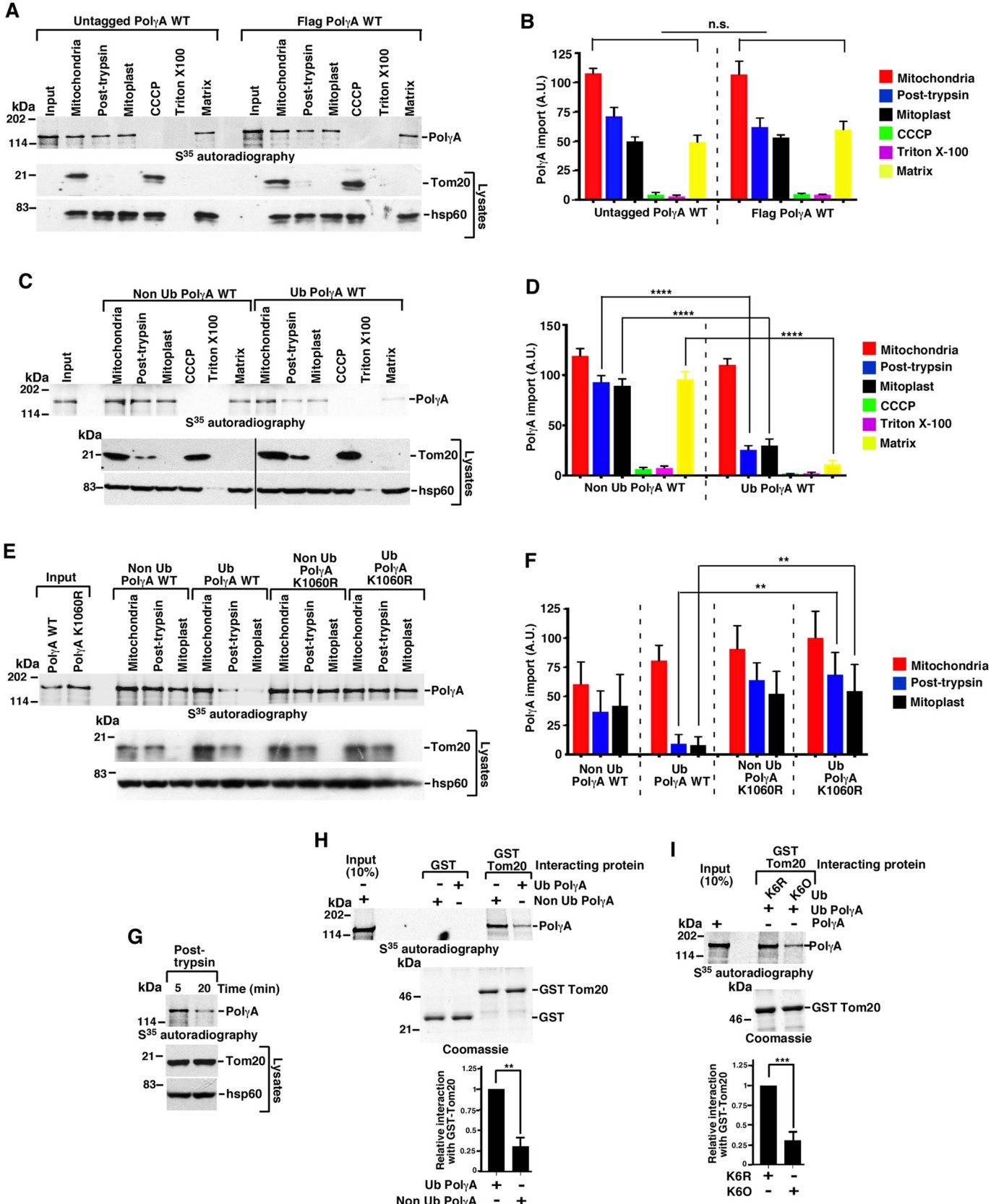

**Fig 3. Ubiquitylation of PolγA negatively regulate its mitochondrial entry. (A, B)** Ubiquitylated Flag-tagged and untagged PolγA enters mitochondria with equal efficiency. **(A)** (Top) Import assays were carried out using the indicated mitochondrial fractions using ubiquitylated S$^{35}$ methionine radiolabeled untagged or Flag-tagged PolγA. The amount of product in each compartment was detected by autoradiography. (Bottom) The integrity of the mitochondria before and after each of the mentioned treatments was determined by carrying out western blot analysis with the indicated antibodies. **(B)** Quantification of (A), done with data from 3 biological replicates. **(C, D)** Ubiquitylated PolγA enters mitochondria with lesser efficiency. **(C)** Same as (A) except ubiquitylated or non-ubiquitylated Flag-tagged PolγA was used. **(D)** Quantification of (C) done with data from 3 biological replicates. **(E, F)** Ubiquitylation site–specific PolγA mutant show enhanced entry into mitochondrial fractions. **(E)** (Top) Same as (A) except the import assay was carried out with Non Ub Flag-tagged PolγA WT, Ub Flag-tagged PolγA WT, Non Ub Flag-tagged PolγA K1060R, and Ub Flag-tagged PolγA. K1060R. **(F)** Quantification of (E), done with data from 3 biological replicates. **(G)** Extent of PolγA ubiquitylation determines its mitochondrial entry. Same as (A) except S$^{35}$ methionine radiolabeled PolγA ubiquitylated by MITOL for either 5 minutes or 20 minutes were used as the substrate. The relative import of PolγA in the post-trypsin fraction was determined by autoradiography. The entire experiment was repeated 3 times, and the same results were obtained. **(H)** Non-ubiquitylated PolγA show enhanced interaction with Tom20. Interaction of S$^{35}$ methionine radiolabeled ubiquitylated PolγA (Ub PolγA) and non-ubiquitylated PolγA (Non Ub PolγA) were carried out with either bound GST or GST Tom20. Post-interaction, the bound radioactivity subjected to SDS-PAGE and detected by autoradiography. Coomassie stained gel shows the levels of bound GST or GST Tom20 used in the interactions. The relative interaction of bound GST-Tom20 with Ub PolγA or Non Ub PolγA has been quantitated from 3 biological replicates. **(I)** K6 ubiquitin linked PolγA show decreased interaction with Tom20. Same as (H) except different types of ubiquitin moieties, namely Ub WT, Ub K6R, or Ub K6O were used in the ubiquitylation reactions for PolγA. Coomassie stained gel shows the levels of bound GST Tom20 used in the interactions. The relative interaction of bound GST-Tom20 with Ub PolγA (generated by using either Ub K6R or Ub K6O) has been quantitated from 3 biological replicates. Numerical values for all graphs can be found in S1 Data. See also S3 Fig. PolγA, polymerase γ subunit A; WT, wild-type.

for details), PEO patients #1 and #2 were degraded to a larger extent upon MITOL overexpression, an effect which was reversed upon MG132 treatment (Fig 4E). PEO mutants #1 and #2 could not enter mitochondria as they interacted to a greater extent with MITOL but did not bind well with Tom20 receptor on the mitochondrial membrane (Fig 4C, bottom panel). In contrast, PEO mutants #3 and #4 bound to Tom20 with better efficiency (Fig 4C, bottom panel; S3D Fig, right panel). Unlike PolγA WT and PEO mutants #3 and #4, expression of PEO mutants #1 and #2 led to decreased incorporation of BrdU within the mtDNA, as observed both by Southwestern and slot blot western analysis (Fig 4F and 4G). Next, we wanted to compare the above *in vivo* results for PolγA-mediated polymerization with that obtained *in vitro* using the corresponding recombinant proteins (S5A Fig). *In vitro* polymerase activity of all the PEO mutants was greatly diminished compared to PolγA WT (S5B and S5C Fig). *In vitro* exonuclease activity of PEO mutants #1, #2, and #3 was also diminished compared to PolγA WT. However, the exonuclease activity of PEO mutant #4 was comparable to that of PolγA WT (as also reported earlier [30]) (S5D and S5E Fig).

To understand whether suboptimal presence or absence of PolγA within the mitochondria affected the repair efficiency of the mtDNA in the PEO patients, a long-range DNA amplification assay based on quantitative polymerase chain reaction (qPCR) was standardized for the entire 16.2-kb mtDNA. If unrepaired bases in the mtDNA were present, they should terminate the polymerase elongation, thereby decrease the output signal during the PCR reaction. The 110-bp mtND1 PCR product was used to normalize for the amount of mtDNA used for each reaction. Results provided evidence that in contrast to cells expressing PolγA WT, the mtDNA repair ability of PEO mutants #1 and #2 was highly compromised (S6A and S6B Fig).

We further wanted to determine the reason behind the inability of some of the PolγA mutants to enter mitochondrial matrix. Since MITOL is known to prefer aggregated proteins as substrates [20–23], we wanted to know whether PEO mutants #1 and #2 were present in the insoluble cellular fraction compared to PolγA WT and PEO mutants #3 and #4. Again, all the proteins were expressed to equivalent levels (S4C Fig). Cell fractionation indicated that PEO patients #1 and #2 were present to a greater extent in the insoluble fraction compared to their corresponding WT counterparts (S6C and S6D Fig).

## Reactivation of PEO mutants

Having demonstrated that PEO mutants #1 and #2 cannot enter mitochondria because of their ubiquitylation by MITOL, we next wanted to determine whether it would be possible to

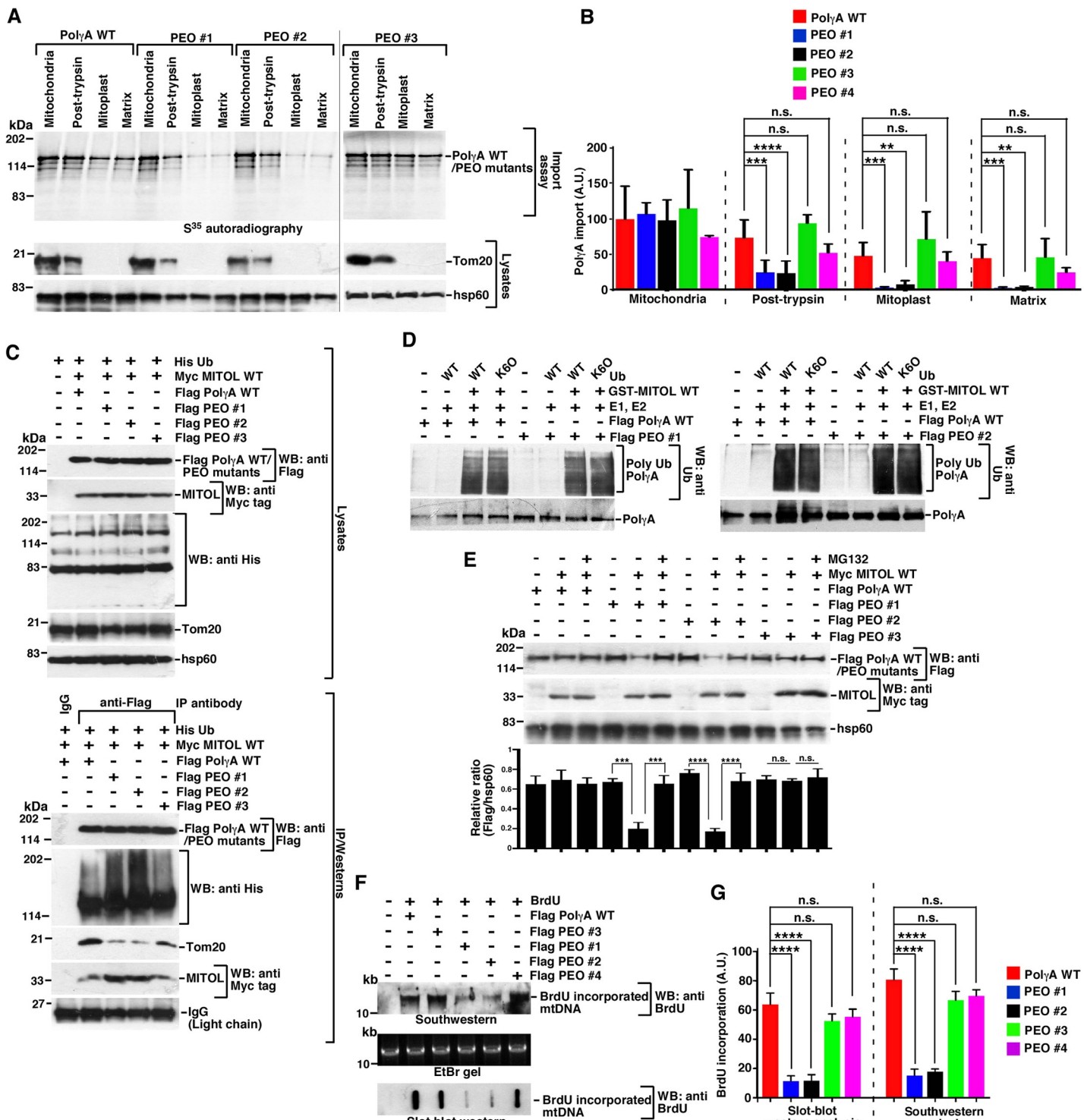

**Fig 4. A subset of PEO mutants has compromised mitochondrial entry. (A, B)** A subset of PEO mutants has compromised mitochondrial entry. (Top) Mitochondrial import assay was carried out using the indicated mitochondrial fractions. S$^{35}$ methionine radiolabeled PolγA WT, PEO mutant #1, PEO mutant #2, and PEO mutant #3 were incubated with each of the mitochondrial fractions. The amount of product in each compartment was detected by autoradiography. (Bottom) The purity of the mitochondrial fractions was determined using the indicated antibodies. The experiment for PEO mutant #3 was done independently, and separate blots were obtained. (B) Quantification of A and S4B Fig is presented. Data are from 3 biological replicates. **(C)** Subset of PEO patients shows enhanced ubiquitylation and reduced binding to Tom20. (Top) Whole cell extracts were made from HEK293T cells transfected with His-Ub, Myc MITOL WT, Flag PolγA WT, Flag PEO mutant #1, Flag PEO mutant #2, and Flag PEO mutant #3. Western blot analysis was carried out with the indicated antibodies. (Bottom) Immunoprecipitations were carried out with anti-

Flag antibody (or the corresponding IgG), and the immunoprecipitates were probed with the indicated antibodies. Three independent biological replicates were carried out, and the same result was obtained. **(D)** MITOL ubiquitylates PEO mutants *in vitro* via K6 linkage. *In vitro* ubiquitylation reactions were carried out using PolγA, Flag PEO mutant #1, Flag PEO mutant #2 as the substrate, MITOL WT, and ubiquitin (WT or K6O). Post-reaction, the products were detected by western blot analysis with the indicated antibodies. Three biological replicates were carried out, and the same results were obtained in each case. **(E)** Subset of PEO mutants has enhanced rate of proteasomal degradation. Whole cell extracts were made from HEK293T cells overexpressing Myc MITOL WT, Flag PolγA WT, Flag PEO mutant #1, Flag PEO mutant #2, and Flag PEO mutant #3 and grown either in absence or presence of MG132. Western blot analysis was carried out with the indicated antibodies. The relative levels of Flag PolγA variants to hsp60 have been quantitated from 3 biological replicates. **(F, G)** Cells expressing subset of PEO mutants have lesser capability of mtDNA replication. (F) mtDNA replication in HEK293T cells expressing Flag PolγA WT, Flag PEO mutant #1, Flag PEO mutant #2, Flag PEO mutant #3, and Flag PEO mutant #4 were determined either by (top) Southwestern analysis or (bottom) Slot blot western using anti-BrdU antibody. (Middle) An EtBr gel for NheI digestion of mtDNA shows equal amount of DNA taken for both the assays. (G) Quantification of (F) from 3 biological replicates. Numerical values for all graphs can be found in S1 Data. See also S4–S6 Figs. His-Ub, His-tagged ubiquitin; IgG, immunoglobulin G; mtDNA, mitochondrial DNA; PEO, progressive external ophthalmoplegia; PolγA, polymerase γ subunit A; WT, wild-type.

reactivate the same PEO mutants. We tested out 2 strategies to reactivate these 2 PolγA mutants that cannot enter mitochondria. In the first strategy, we wanted to check whether upon ablation of MITOL, PolγA mutants #1 and #2 become reactivated. Indeed, ablation of MITOL in HeLa shMITOL cells led to entry of both PEO mutants #1 and #2 into mitoplast (Fig 5A). Next, we wanted to determine the extent of ubiquitylation of these 2 mutants in comparison to PolγA WT in the whole mitochondrial lysates (Fig 5B, left). As expected, the extent of ubiquitylation in PEO mutants #1 and #2 was much higher than PolγA WT in HeLa shGFP cells (Fig 5B, middle). However, in HeLa shMITOL cells, low but equivalent levels of ubiquitylation were observed for all the 3 PolγA variants (Fig 5B, right), indicating that the lack of MITOL was the critical factor for their successful mitochondrial entry into the mitochondria. Both Southwestern and slot blot analysis indicated that HeLa shMITOL cells expressing PolγA WT or PEO mutant #1 or #2 were able to incorporate BrdU to significant levels, thereby indicating successful reactivation for these mutants (Fig 5C and 5D).

However, we recognized that given the importance of MITOL as an E3 ligase and its many of substrates having critical cellular functions [31], inactivating it may not be a viable solution to reactivate PEO mutants. Hence, as a second strategy, we wanted to determine whether it is possible to increase the entry of PEO mutants #1 and #2 into mitochondrial matrix and thereby attain their reactivation. We hypothesized that since PEO mutants #1 and #2 were hyperubiquitylated, changing lysine 1060 residue to arginine (PolγA K1060R) should abolish their ubiquitylation by MITOL and thereby help them to bind to Tom20 and allow them to enter the mitochondrial matrix. Immunoprecipitation experiments revealed that K1060R counterparts of both PEO mutants #1 and #2 were not hyperubiquitylated (Fig 6A, right panel). This allowed PEO mutant #1 K1060R and PEO mutant #2 K1060R to bind as strongly with Tom20 as PolγA WT (Fig 6A, right panel). The K1060R counterparts of the PEO mutants #1 and #2 entered the mitochondria with efficiencies which were significantly more than that observed for PEO mutants #1 and #2 themselves (Fig 6B and 6C). Slot blot western and Southwestern analysis also revealed greater incorporation of BrdU in cells expressing the K1060R versions of mutants #1 and #2 (Fig 6D and 6E), indicating that the lack of mtDNA replication in PEO mutants #1 and #2 can be reverted to an extent, and significant reactivation of these 2 PEO mutants have been achieved. Finally, we wanted to determine whether this reactivation was due to decrease in the accumulation of PEO mutants #1 and #2 K1060R variants in the insoluble fraction. Compared to PEO mutants #1 and #2, their K1060R variant accumulated much less in the insoluble fraction (S6E and S6F Fig), which possibly made them less likely to become MITOL substrates.

## Discussion

Studies have been shown that MITOL carries out its function by acting as a multifunctional protein on the OMM. MITOL acts as a key signaling factor which controls the mitochondrial

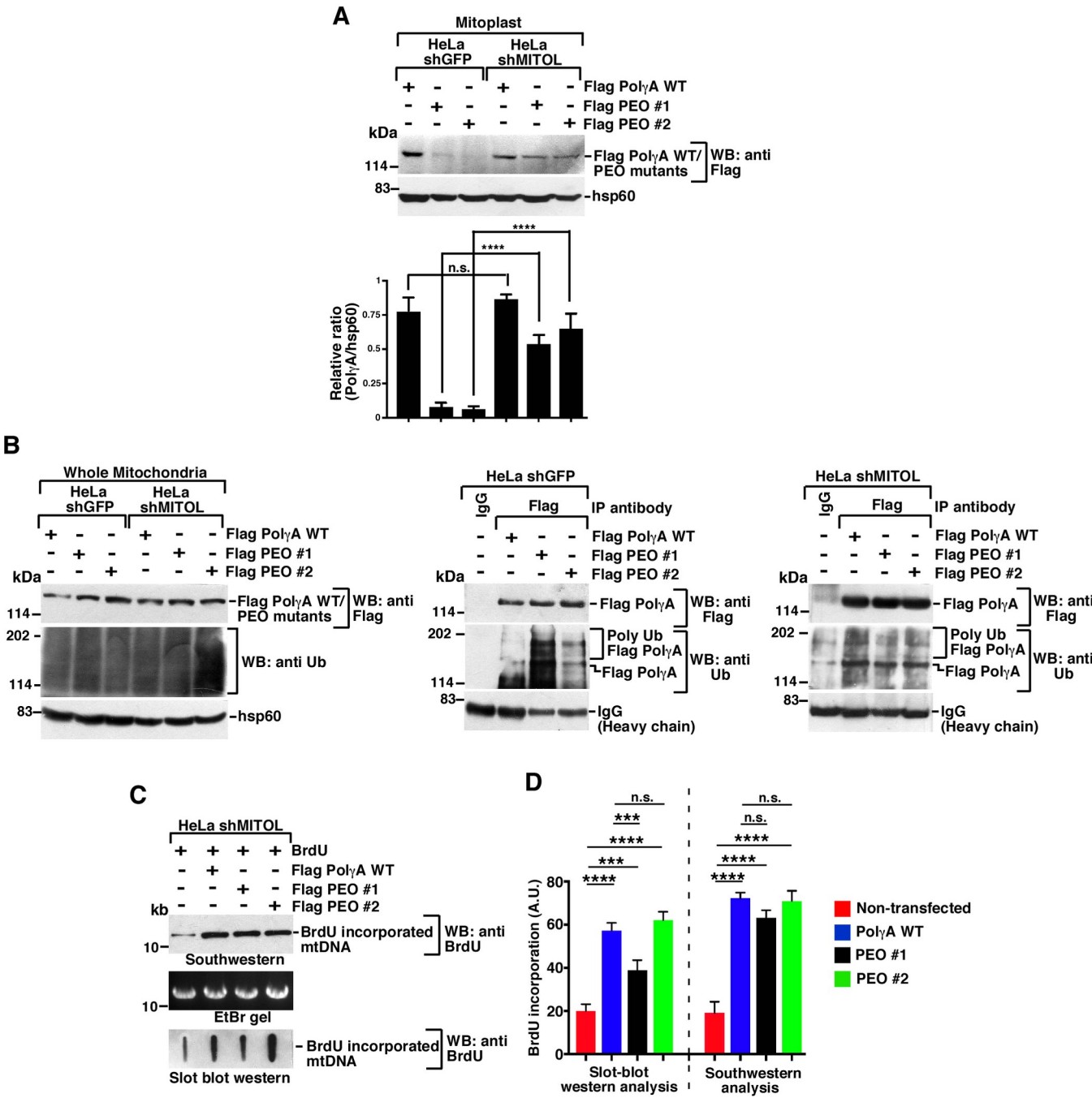

**Fig 5. PEO mutants can be reactivated by depleting MITOL. (A)** HeLa shMITOL cells show increased entry of PEO mutants. Mitoplasts were isolated from HeLa shGFP and HeLa shMITOL cells expressing Flag PolγA WT, Flag PEO mutant #1, and Flag PEO mutant #2. Western blot analysis was carried out with the indicated antibodies. The relative levels of PolγA to hsp60 have been quantitated from 3 biological replicates. **(B)** PEO mutants undergo the same level of ubiquitylation as PolγA WT in mitochondrial lysates from HeLa shMITOL. (Left) Whole mitochondrial lysates were isolated from HeLa shGFP and HeLa shMITOL cells expressing Flag PolγA WT, Flag PEO mutant #1, and Flag PEO mutant #2. Western blot analysis was carried out with the indicated antibodies. (Center and right) Immunoprecipitations were carried out with anti-Flag antibody (or the corresponding IgG) using whole mitochondrial lysates isolated from HeLa shGFP and HeLa shMITOL cells expressing Flag PolγA WT, Flag PEO mutant #1, and Flag PEO mutant #2. The immunoprecipitates were probed with the indicated antibodies. Three biological replicates were carried out, and the same results were obtained in each case. **(C, D)** Lack of MITOL allows PEO mutants to carry out mtDNA replication. (C) mtDNA replication in HeLa shMITOL cells expressing Flag PolγA WT, Flag PEO mutant #1, and Flag PEO mutant #2 were determined either by (top) Southwestern analysis or (bottom) Slot blot western using anti-BrdU antibody. (Middle) An EtBr gel for NheI digestion of mtDNA shows equal amount of DNA taken for both the assays. (D) Quantification of (C) from 3 biological replicates. Numerical values for all graphs can be found in S1 Data. IgG, immunoglobulin G; mtDNA, mitochondrial DNA; PEO, progressive external ophthalmoplegia; WT, wild-type.

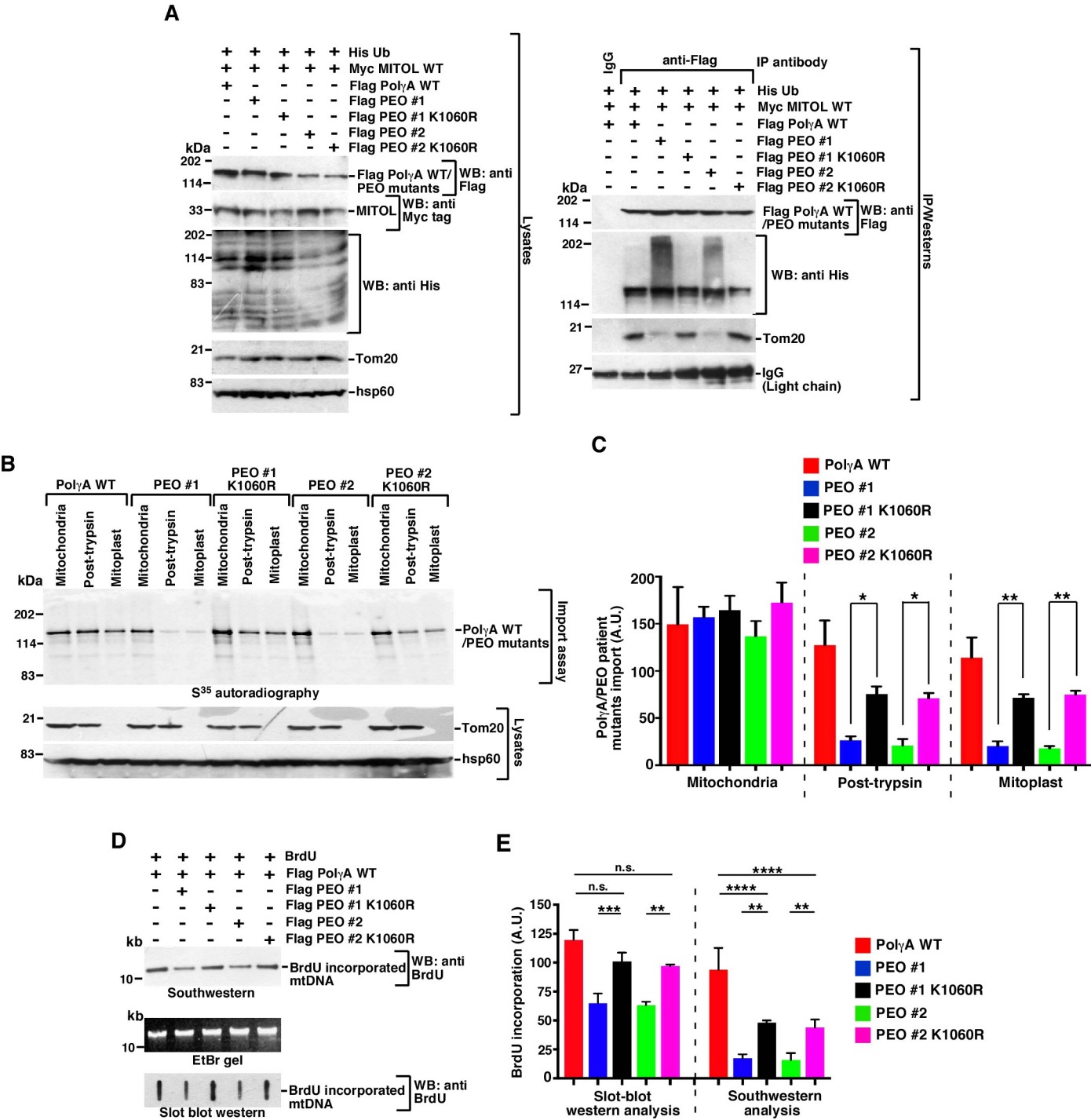

**Fig 6. PEO mutants can be reactivated by preventing their ubiquitylation at K1060.** (A) A subset of PEO mutants can be reactivated by mutating their site of ubiquitylation by MITOL. (Left) Whole cell extracts were made from HEK293T cells transfected with Myc MITOL WT, Flag PolγA WT, Flag PEO mutant #1, Flag PEO mutant #1 K1060R, Flag PEO mutant #2, and Flag PEO mutant #2 K1060R. Western blot analysis was carried out with the indicated antibodies. (Right) Immunoprecipitations were carried out with anti-Flag antibody (or the corresponding IgG), and the immunoprecipitates were probed with the indicated antibodies. Three independent biological replicates were carried out, and the same result was obtained. **(B, C)** K1060R counterparts of PEO mutants #1 and #2 can enter mitochondria with similar efficiency was PolγA WT. (B) (Top) Mitochondrial import assay was carried out using the indicated mitochondrial fractions. $S^{35}$ methionine radiolabeled Flag PEO mutant #1, Flag PEO mutant #1 K1060R, Flag PEO mutant #2, and Flag PEO mutant #2 K1060R were incubated with each of the mitochondrial fractions. The amount of product in each compartment was detected by autoradiography. (Bottom) The purity of the mitochondrial fractions was determined using the indicated antibodies. (C) Quantification of (B) from 3 biological replicates. **(D, E)** Cells expressing K1060R variant PEO mutants #1 and #2 can be reactivated to incorporate BrdU. (D) mtDNA replication in HEK293T cells expressing Flag PolγA WT, Flag PEO mutant #1, Flag PEO mutant #1 K1060R, Flag

PEO mutant #2, and Flag PEO mutant #2 K1060R were determined either by (top) Southwestern analysis or (bottom) Slot blot western using anti-BrdU antibody. (Middle) An EtBr gel for NheI digestion of mtDNA shows equal amount of DNA taken for the 2 assays. (E) Quantification of (D) from 3 biological replicates. Numerical values for all graphs can be found in S1 Data. See also S6 Fig. His-Ub, His-tagged ubiquitin; IgG, immunoglobulin G; PEO, progressive external ophthalmoplegia; WT, wild type.

dynamics and interconnectivity between the mitochondria and endoplasmic reticulum [24,32–34]. Perhaps more importantly, MITOL also acts as a component in the mitochondrial quality control at the OMM by ubiquitylating and degrading diverse toxicity inducing proteins like mutant SOD1, mutant short chain acyl CoA dehydrogenase, mutant MITOL itself and polyglutamine expanded protein, and MAVS aggregates [20–23].

Very recently, it has been shown that MITOL-mediated ubiquitylation of mitochondrial proteins negatively regulates their entry into the organelle [28]. This can be rationalized as mitochondria needs a level of control which allows the entry of these proteins in a regulated and "on demand" manner. We now provide evidence to show that compared to PolγA ubiquitylated by MITOL, non-ubiquitylated PolγA binds better with Tom20, which allows it to enter the mitochondria, and thereby carry out mtDNA replication. Hence, we conclude that the mere presence of MLS is not sufficient for proteins to enter into the mitochondrial matrix, and a finer regulatory control like ubiquitylation is necessary.

Previous reports have suggested that ubiquitylation by MITOL of misfolded proteins and their subsequent elimination from the system is important for optimal mitochondrial homeostasis [20–23]. We now provide evidence that 50% of the examined PEO mutants undergo hyperubiquitylation by MITOL on the OMM, which prevents them from binding to Tom20 receptor with maximum efficiency. Thus, ubiquitylation of these PEO patient-derived PolγA diminishes their entry into the mitochondrial matrix and consequently affects their functions in mitochondrial replication and mtDNA repair. Further, PEO mutants #1 and #2 were present much more in the insoluble fraction compared to their WT counterpart. It is conceivable that these PEO mutants which cannot enter mitochondria possibly have alterations in their respective conformations after MITOL-dependent ubiquitylation, which allows them to become better MITOL targets. In fact, it has been demonstrated that conformation change in microtubule-associated protein 1B light chain 1 (LC1) allowed it to be targeted by MITOL [21].

MITOL has been shown to ubiquitylate its substrates either via K48 or K63 linkages [20,23,26]. Here, we demonstrate that MITOL can ubiquitylate PolγA via noncanonical K6 linkage. Usage of "R" and "O" ubiquitin mutants *in vitro* and K6 affimer *in vivo* provided the experimental evidence for this specific linkage. Interestingly, it has been recently shown that Mitofusin2, which is targeted by MITOL [24], is also targeted by another E3 ligase HUWE1 in a K6-linked manner [27]. K6-linked ubiquitylation linkages accumulate with faster kinetics after proteasomal inhibition [35]. This is possibly due to its reduced rates of USP14-dependent deubiquitylation of K6 linkages due to their structural orientation [36]. However, we provide evidence that PolγA is not a substrate of HUWE1. Instead, PolγA is specifically ubiquitylated by MITOL via K6 linkage.

Evidence exists detailing the biochemical mechanisms by which the various mutant PolγA proteins function [37,38]. It is undeniable that the compromised catalytic activity of PEO mutants play a major role in mtDNA replication defect in these patients. For example, active site PolγA Y955C mutant (PEO mutant #4 in our study) has been shown to have dominant effect leading to less than 1% polymerase activity, decreased processivity, increased error prone DNA synthesis, and stalling phenotypes [2,39,40]. The spacer domain mutation PolγA A467T (PEO mutant #2 in our study) has compromised catalytic efficiency and decreased

polymerase activity, DNA binding, and processivity—effects which have been linked to its lack of optimal interactions with the accessory factor [3]. The most common PolγA mutation W748S (PEO mutant #3 in our study) also show significantly reduced polymerase activity [4]. In this study, the polymerization and exonuclease assays performed using recombinant PolγA WT and PEO mutants have also validated the published biochemical work. Furthermore, structural studies on WT PolγA [25] have validated most of the biochemical published reports. However, it should be noted that the biochemical or structural studies were all carried out with recombinant purified proteins and not in an *in vivo* milieu, only where posttranslational modifications like MITOL-dependent ubiquitylation on PolγA can occur.

Similarly, lack of one to one correlation between the *in vitro* and *in vivo* results has been noticed in some of our other results. For example, while we observe that USP30 deubiquitylates PolγA *in vitro*, its expression in asynchronously growing cells does not rescue MITOL-dependent decrease in the level PolγA nor alter the extent of PolγA ubiquitylation. Based on our data and also the recent literature [28], we believe that *in vivo* USP30 may act as a deubiquitylase to MITOL-ubiquitylated PolγA only under very specific physiological conditions, which needs to be further investigated.

In recent time, efforts are being made to carry out mitochondrial genome engineering using multiple manipulative techniques [41,42]. Ablation of MITOL can allow the 2 PEO mutants #1 and #2 to enter mitoplast and be reactivated. However, removal of MITOL from cellular milieu will have substantial effects on vital cellular functions and will be difficult to implement. Hence, the alternate usage of K1060R derivatives of PEO mutants #1 and #2 will ensure lack of MITOL-specific polyubiquitylation and allow their better binding to Tom20, leading to the reentry of these PolγA variants into mitochondria and become functionally active. It will be interesting to test whether the reactivation of PolγA can actually be carried out *in vivo* using mitochondrial genome editing techniques.

In conclusion, we have deciphered the mechanism by which PolγA enters into the mitochondria to an optimal extent (Fig 7). We show that for a subset of the PolγA mutants, apart from the lack of their optimal catalytic activity, an additional level of regulation exists within the cells which negatively control their mitochondrial entry and thereby their functions inside the mitochondrial matrix. These results complement the current dogma that MLS of Polγ and its interactions with TOM complex are sufficient for its entry into mitochondria. It is interesting to note that under pathological abnormalities like during certain cases of PEO, this fined-tuned mechanism of WT PolγA import is no longer operational. Hence, under these conditions where hyperubiquitylation of PEO mutants occur, the reactivation of PolγA may help in reverting phenotypes associated with mitochondrial disorders and thereby lead to near optimal levels of mtDNA replication.

## Materials and methods

### Reagents

All the antibodies and recombinant DNAs used are listed in S2 and S3 Tables, respectively. All chemicals, recombinant proteins, cell lines, siRNAs, and kits used are listed in S4 Table. Primers used for different assays are listed in S5 Table.

### Plasmids, siRNAs, and RT-qPCR

All mutants listed in S3 Table were generated by site directed mutagenesis. Catalytically dead MITOL (MITOL CD) was generated by incorporating the following changes in the WT amino acid sequence: C65S, C68S, and H43S. GST MITOL WT/CD was generated by sub-cloning MITOL WT sequence at BamH1/XhoI sites in pGEX4T-1. Untagged PolγA, named as pcDNA

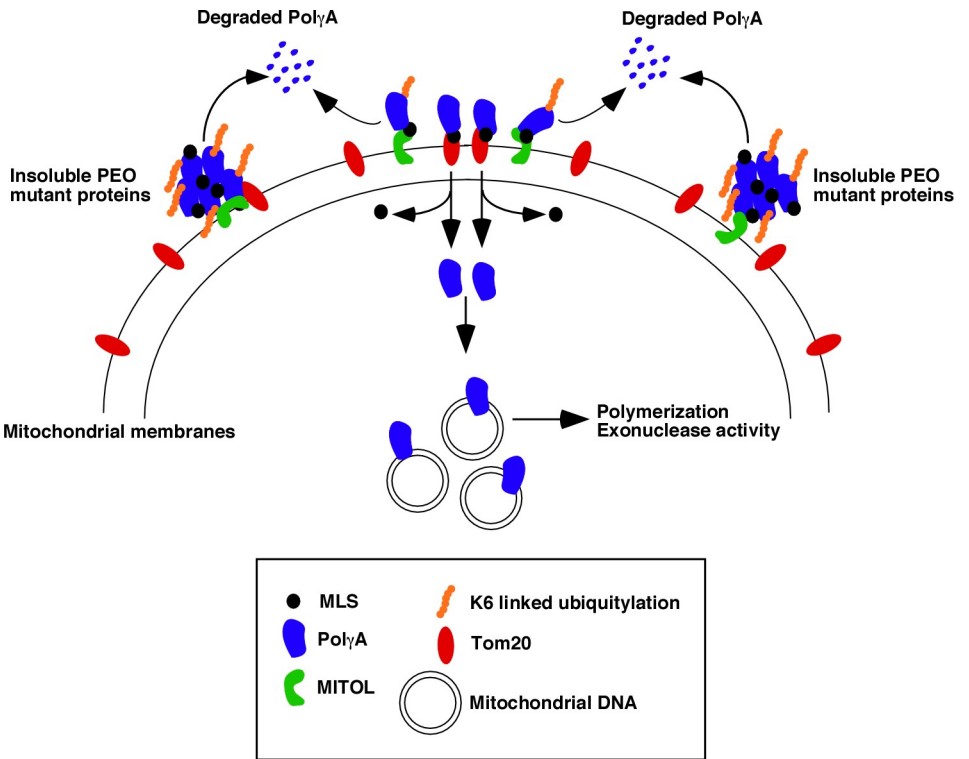

**Fig 7. MITOL-dependent ubiquitylation negatively regulates PolγA entry into mitochondria and thereby mtDNA replication.** PolγA has a key role in mtDNA replication. WT PolγA is targeted by MITOL, an E3 ligase present in the OMM. MITOL ubiquitylates PolγA at K1060 via K6 linkage. This ubiquitylation event prevents PolγA from efficiently binding to Tom20. Consequently, only non-ubiquitylated PolγA (which binds with Tom20) could enter into the mitochondrial matrix, thereby allowing it to do its functions during mtDNA replication. This regulated entry of PolγA into mitochondria is no longer operational under pathological situations. In cells expressing certain PolγA variants present in PEO patients, the mutant PolγA proteins are hyperubiquitylated by MITOL and hence cannot enter into mitochondria. Instead, these proteins are present in the insoluble fraction, which allows them to be recognized and targeted for degradation. Consequently, the lack of entry of the mutated PolγA proteins into mitochondria leads to decreased mtDNA replication and thereby diminished ability to maintain the mitochondrial genome integrity. MLS, mitochondrial localization signal; mtDNA, mitochondrial DNA; PEO, progressive external ophthalmoplegia; PolγA, polymerase γ subunit A; WT, wild-type.

3.1 hygro (+) PolγA WT, was generated by sub-cloning PolγA WT sequence at NheI/BamH1 sites in pcDNA 3.1 hygro (+). The siRNA sequences (5′ to 3′) for MITOL were siRNA#1: GCU CUA UCU AUU GGA CAG and siRNA#2: UCU UGG GUG GAA UUG CGU U [43]. The siRNA sequence (5′ to 3′) to HUWE1 was CCC GCA UGA UCU UGA AUU U [44]. RNA isolation for reverse transcription quantitative polymerase chain reaction (RT-qPCR) and its conversion to cDNA were carried out using Trizol reagent (Thermo Fisher Scientific, Waltham, Massachusetts, United States) and Reverse transcriptase core kit (Eurogentec, Liege, Belgium), respectively, according to the manufacturers' protocols. The RT-qPCR reactions were carried out in QuantStudio 3 Real-time PCR system (Thermo Fisher Scientific).

## Cell culture treatments and confocal imaging

Both plasmid and siRNA transfections were carried out with Lipofectamine 2000 (Thermo Fisher Scientific) for 6 hours according to manufacturer's protocol. In a 6 well cluster, either 0.25 to 1 μg of the plasmid DNA or 200 pmoles of the respective siRNAs or siControl were used. For all cell culture experiments (except in Fig 4E), lysates were made 24 hours post-transfection. To determine the difference in the rate of ubiquitylation and degradation between WT

and mutant PolγA (as seen in Fig 4E), lysates were made 12 hours post-transfection. This included the period of MG132 (10 μM) treatment which was carried out for 5 hours immediately prior to the preparation of the lysates. CHX (1 mM) treatments were carried out for the indicated time periods. To obtain the soluble and insoluble fractions, the transfected cells were lysed in a lysis buffer (50 mM Tris-HCl pH 7.4, 150 mM NaCl, 5 mM EDTA, 1% Triton X-100, and protease inhibitor cocktail) by pipetting gently and rotating for 10 minutes at 4˚C. Soluble and insoluble fractions were separated by centrifugation at 20,000g for 20 minutes at 4˚C. Immunofluorescence staining was carried out as described [45,46]. All imaging was carried out in LSM 510 Meta System (Carl Zeiss, Germany) with 63×/1.4 oil immersion objective. The laser lines used were Argon 458/477/488/514 nm (for FITC), DPSS 561 nm (for Alexa Fluor 594), and a Chameleon Ultra auto-tunable femtosecond laser 690–1050 nm (for DAPI). All parameters were kept constant during image acquisitions using LSM5 software for particular experiments. Colocalization factor is defined as [(Fraction of cells having colocalization) × (Fraction of foci colocalized per cell)] × 100 [47]. Colocalization factor was calculated for 100 cells spread over 3 independent experiments.

### *In vitro* ubiquitination assay

During *in vitro* ubiquitylation assays, GST MITOL WT/CD (1.6 μM) were incubated with equal amount (2 μl) of $S^{35}$ methionine labeled *in vitro* transcribed and translated PolγA in the ubiquitylation buffer (40 mM Tris-HCl, pH 7.5, 5 mM MgCl$_2$, and 2 mM ATP) in presence of 100 ng E1 (UBE1), 300 ng E2 (UbcH5a), and 2 μM Ubiquitin. MITOL WT/CD (0.8 μM) was added at the end to initiate the reaction. Normally, the ubiquitylation reactions were carried out at 30 °C for 2 hours. To obtain gradient of PolγA ubiquitylation, the reactions were stopped after 5 minutes or 20 minutes. Post-reaction, samples were boiled with SDS-PAGE sample buffer and subjected to western analysis. It is to be noted that PolγA ubiquitylated smears were not observed upon autoradiography. This was due to the fact that the SDS-PAGE gels had to be subjected to fluorography (as $S^{35}$ Methionine is a weak beta emitter), and the gels dried at 65˚C for 3 to 4 hours, before being exposed for autoradiography. The process of fluorography and/or the heating the gels during the drying process degraded the ubiquitylated products for a high molecular weight substrate like PolγA.

Apart from WT ubiquitin, "R" and "O" ubiquitin mutants have been used in these assays. In "R" ubiquitin mutants, only a particular lysine residue in ubiquitin is mutated, while in "O" ubiquitin mutants, only 1 lysine is present on ubiquitin, and all other 6 lysines are mutated. For all the linkage studies a limiting concentration (250 nM) of WT ubiquitin, "R" and "O" ubiquitin mutants were used.

### *In vitro* deubiquitination assay

The assay was carried out according to [48]. MITOL-dependent ubiquitylation was standardized in a ubiquitylation–deubiquitylation buffer (20 mM Tris-Cl pH 8.0, 0.01% Triton X-100, 1 mM L-glutathione, 0.03% BSA, 5 mM MgCl2, and 2 mM ATP). The ubiquitylation reaction was done for 90 minutes at 30˚C in presence of 100 ng E1 (UBE1), 300 ng (UbcH5a), 2 μM Ubiquitin, and 0.8 μM MITOL. USP30 (0.8 μM) was added either with MITOL (called simultaneous reaction) or after the completion of the MITOL-dependent ubiquitylation (called sequential reaction). For the sequential reaction, the incubation was continued for a further 90 minutes.

### Isolation of mitochondria

Mitochondria were isolated from both HEK293T cell lines. For this purpose, 5 to $10 \times 10^9$ HEK-293T cells were grown as monolayers, washed with 1X PBS, and harvested in

mitochondria isolation buffer (20 mM Hepes, pH 7.5, 1.5 mM MgCl$_2$, 1 mM EDTA, 1 mM EGTA, 210 mM sucrose, and 70 mM mannitol). HEK293T cell suspension was subjected to 60 gentle strokes using Dounce homogenizer. The supernatant obtained was again centrifuged at 10,000Xg for 15 minutes at 4˚C. The resultant mitochondrial pellet was washed twice and suspended in 200 μl buffer containing 250 mM sucrose, 5 mM magnesium acetate, 40 mM potassium acetate, 10 mM sodium succinate, 1 mM DTT, and 20 mM Hepes/KOH, pH 7.4.

## *In vitro* transcription, translation, and import assay

*In vitro* transcription and translation was carried out with 1 μg of the supercoiled DNA in presence of $^{35}$S methionine using T7 Quick Coupled Transcription/translation system. The reaction was done for 90 minutes at 30˚C, following which products were checked by autoradiography in SDS-PAGE. The labeled WT PolγA and PEO mutants were used in import assays as described previously [46]. Six such reactions were set up in parallel with the mitochondria isolated from HEK293T cells. In the first reaction, labeled proteins (2 μl), which have been either ubiquitylated or non-ubiquitylated by MITOL, were incubated with 200 μg of isolated total mitochondria in import buffer (0.25 M sucrose, 1.5 mM MgCl$_2$, 2.5 mg/ml BSA, and 10 mM HEPES, pH 7.2) supplemented with 2 mM ATP, 2 mM GTP, 5 mM Magnesium Acetate, 20 mM KCl, and 2 mM succinate. Reaction was initiated at 37˚C by ATP and lasted for 60 minutes. For this first reaction, one-half of each sample was assessed for the presence of the proteins by western analysis in the total mitochondria, and the other half was kept for autoradiography. For the second reaction, the import assay was carried out as above. Further, the mitochondria were treated with trypsin (final concentration 20 μg/ml) on ice for 15 minutes to remove the outer membrane proteins. After inhibiting trypsin with soybean trypsin inhibitor (final concentration 50 μg/ml), mitochondria were re-isolated by passing it through a sucrose cushion (0.8 M sucrose in 10 mM HEPES, pH 7.2) by centrifuging at 12,000 rpm for 10 minutes (to remove the proteins which have been digested by trypsin treatment). At this stage, half of this second reaction was kept for western and the other half for autoradiography. In the third reaction, the reaction was done as above. Further, the mitoplast was isolated by resuspending the mitochondria in a hypotonic buffer (20 mM KCl and 10 mM HEPES, pH 7.2) or treated with 0.1% digitonin and incubated on ice for 20 minutes. The resulting mitoplasts were re-isolated by centrifugation at 12,000 rpm for 10 minutes and passed through the sucrose cushion to remove the OMM proteins. At this stage of the third reaction, half of the material was kept for western and the other half for autoradiography. In the fourth reaction, after removal of the OMM proteins, the mitochondrial matrix was isolated by resuspending the samples in a hypotonic buffer (5 mM Tris-HCl 7.4 and 1 mM EDTA) and sonicated mildly (2 pulse, 10 seconds each with 10 seconds off between the cycles). Sonicated samples were centrifuged at 40,000 rpm for 45 minutes at 4 ˚C. The resultant pellet constituted the inter-mitochondrial membrane (IMM), and supernatant contained the mitochondrial matrix fraction. The supernatant was precipitated using Trichloroacetic acid (TCA) (1 volume of 100% TCA to 4 volumes of supernatant) to concentrate the proteins. Again, the sample were divided in 2 parts—half of this reaction was kept for western and the other half for autoradiography. In fifth reaction, import assay was carried out in presence of 20 μM CCCP to dissipate membrane potential. Further, the mitochondria were treated with trypsin (final concentration 20 μg/ml) on ice for 15 minutes to remove the outer membrane proteins. After inhibiting trypsin with soybean trypsin inhibitor (final concentration 50 μg/ml), mitochondria were re-isolated by passing it through a sucrose cushion (0.8 M sucrose in 10mM HEPES, pH 7.2) by centrifuging at 12,000 rpm for 10 minutes (to remove the proteins which have been digested by trypsin treatment). At this stage, half of this reaction was kept for western and the other half for

autoradiography. In sixth reaction, after import assay, mitochondria were lysed by treatment with 1% Triton X-100 (V/V) before the trypsin treatment. Mitochondria were re-isolated by passing through sucrose cushion. Further samples were divided into two half, one for western blot and the other half for autoradiography.

## Protein purification and interaction studies

All proteins in *Escherichia coli* were expressed in BL21-CodonPlus-RP competent cells strain by adding 1 mM of Isopropyl-1-thio-β-D-galactopyranoside (IPTG) and incubating the culture at 200 rpm, 18°C for 6 hours. The sonication was carried out in Bandelin Sonoplus HD2070 sonicator using a MS73 probe. Cell slurry was made in 1X PBS supplemented with Protease inhibitor cocktail, 1 M phenylmethylsulfonyl fluoride (PMSF, final concentration 1 mM) and 1 M dithiothreitol (DTT, final concentration 1 mM). A minimum of 5 sonication cycles (30 seconds on, 30 seconds off, 50% cycle, and 20% power) was used. Post sonication, Triton X-100 (prepared in 1X PBS) was added to the suspension at a final concentration of 2%. The suspension was kept for mixing in end-to-end rocker for 30 minutes at 4°C, followed by centrifugation for 30 minutes at 12,000 rpm at 4°C. Subsequently, the supernatant was bound to glutathione–sepharose resin equilibrated in GST buffer (50 mM Tris-HCl (pH 7.5), 100 mM KCl, 10 mM MgCl$_2$, 5% Glycerol, and 0.5% NP-40). The bound proteins were washed thrice with GST buffer, loaded onto Poly-Prep Chromatography column kept at 4°C, and eluted by using a stepwise gradient of glutathione (5 mM and 10 mM prepared in GST buffer). The samples from the eluted fractions were checked in a Coomassie stained gel. The fractions containing the pure proteins were pooled and dialysed against the dialysis buffer (25 mM Tris pH 7.5, 140 mM NaCl, 15% Glycerol, and 1 mM DTT) overnight with 2 changes. The dialyzed purified proteins were then quantified, aliquoted in a storage buffer (25% glycerol, 25 mM Tris pH 7.5, and 140 mM NaCl) and stored in −80°C till further use. To prepare purified proteins in mammalian cells, Flag or Myc-tagged proteins were over expressed in HEK 293T cells for 24 hours. Lysates were prepared from HEK 293T using M2 lysis buffer and expression checked using anti-Flag or anti-Myc tag antibodies. Post-checking, Flag or Myc-tag proteins were immuno-purified using either anti-Flag beads or anti-Myc tag antibodies.

For anti-Flag or anti-Myc tag immunoprecipitations, 100 to 200 μg of lysates overexpressing the respective proteins was used, while for endogenous interactions, 500 μg to 1 mg of the mitochondrial extracts were taken. Using anti-Flag beads (1 to 2 μl) or the antigen-specific antibodies (1 to 2 μg), immunoprecipitations were carried out for 4 hours at 4°C, after which the complexes were washed thrice with the respective lysis buffers at 4°C. For elution of Flag bound proteins, the protein bound beads were incubated with Flag peptide (final concentration 100 μg/ml) for 1 hour. The supernatant obtained post-centrifugation was checked by western blot analysis and Coomassie. To determine the interacting proteins, the immunoprecipitates were subjected to SDS-PAGE and probed with the corresponding antibodies. For *in vitro/in vivo* PolγA-MITOL interactions, GST-tagged proteins (1 to 2 μg) were incubated with the lysates (100 to 200 μg) overexpressing the corresponding proteins. For interactions between PolγA with GST-MITOL, 1 to 2μl of [35S] Methionine radiolabeled *in vitro* transcribed and translated PolγA was incubated with the bound GST proteins (1 to 2 μg) for 6 hours. The bound radioactivity was detected by autoradiography.

## Immunoblotting

For western blotting, lysate was prepared using RIPA buffer [1 mM Tris-HCl pH 7.8, 150 mM NaCl, 2% Triton X-100, 1% (w/v) Sodium deoxycholate, and 0.1% (w/v) SDS supplemented with 1X PIC and 1 mM PMSF]. For western blotting, equal amounts (typically 40 to 50 μg) of

the lysates were run in appropriate percentage of SDS-PAGE gels, transferred to nitrocellulose membranes (usually overnight at 40 mA per gel), and incubated with the respective primary antibodies (listed in S2 Table) overnight at 4˚C with gentle rocking. Next day, the blots were incubated with the respective secondary antibodies for 1 hour and washed and developed using the chemiluminescent detection system.

### Primer extension assay

Primer extension assays were carried out according to a published protocol [49,50]. The primer-template pair (S5 Table) comprised of the first 40 nucleotides of the mtDNA replication origin. The annealed substrate (10 μM) was incubated with the indicated amounts of the PolγA/B2 holoenzyme in a buffer containing 10 mM Tris-HCl, pH 7.4, 8 mM MgCl2, 0.1 mM DTT, 100 μg BSA/ml, 50 μM each of dCTP, dGTP, and dTTP, 1 μM dATP, and [α-P$^{32}$] dATP (0.5μCi). The products were analyzed in a 15% acrylamide-8M urea gel. Primer extension was visualized by the incorporation of [α-P$^{32}$] dATP due to the polymerization functions of PolγA/B2. The radioactive 40-mer produced was quantitated in a phosphoimager.

### Exonuclease assay

Exonuclease assay was carried out as previously described [50,51]. The [γ-$^{32}$P] ATP labeled 25-mer primer with 4 mismatches at 3′ end was annealed to 45-mer template (S5 Table). Annealed substrate (10 nM) was incubated with the indicated amounts of PolγA in a buffer containing 25 mM HEPES-KOH, pH 7.6, 1 mM βME, 5 mM MgCl$_2$, 0.05 ng/ml BSA, and 10 mM NaCl. The reaction was initiated by adding PolγA and incubated at 37 $^{o}$C for 3 minutes. The products were analyzed in a 15% acrylamide-8M urea sequencing gel. Exonuclease activity of PolγA was visualized by the formation of progressively smaller products due to the exonuclease functions of PolγA. All smaller products were quantitated in a phosphoimager.

### Slot blot and Southwestern analysis

HEK293T cells were transfected the PolγA variants. After 24 hours, cells were treated with 10 μM BrdU and incubated for another 24 hours. Transfected cells were harvested, and mtDNA was isolated using mtDNA isolation kit (BioVision, Milpitas, California, US), according to manufacturer's protocol. For slot blot analysis, mtDNA (2 μg) were boiled at 100˚C for 10 minutes and transferred directly onto nitrocellulose membrane using slot blot apparatus (Cleaver Scientific, England). For Southwestern analysis, 1 μg of the isolated mtDNA's were digested with NheI restriction enzyme and separated on 1% agarose gel. DNA were transferred onto nitrocellulose membrane using VacuGene XL (GE Healthcare, Chicago, Illinois, US) according to manufacturer's protocol. Post-transfer, membranes obtained were UV cross-linked, blocked with 100% normal horse serum for 1 hour at room temperature, and probed using anti-BrdU antibody to check BrdU incorporation.

### Long-range mtDNA amplification assay

A total of 20 ng of mtDNA template was used to amplify mtND1 and full-length mtDNA using LongAmp Taq DNA polymerase using the primers whose sequence is in S5 Table. PCR reactions were performed at 94˚C for 30 seconds followed by 30 cycles at 94˚C for 30 seconds, 65˚C for 14 minutes, and a final elongation for 10 minutes. The products were separated by electrophoresis on a 1% agarose gel stained with ethidium bromide.

## Statistical analysis

All data are presented as mean ± SD. * $p < 0.05$, ** $p < 0.01$, *** $p < 0.001$, **** $p < 0.001$, while n.s. indicates that the result is not significant. The statistical analyses employed for every experiment are shown in S6 Table.

## Supporting information

**S1 Fig. Factors determining the interaction between MITOL and PolγA (related to Fig 1).** **(A, B) Overexpression and ablation of MITOL does not alter PolγA transcript levels.** HEK293T cells were either (A) transfected with either Myc MITOL or the corresponding vector or (B) transfected with either siControl or siMITOL RNA was isolated and RT-qPCR carried out to detect the levels of MITOL and PolγA. The transcript levels of GAPDH was used as control. **(C) Endogenous PolγA colocalize with MITOL.** Asynchronously growing NHFs were stained with antibodies against PolγA and MITOL. DNA was stained by DAPI. Scale, 5 μM. The colocalization factor has been indicated. Representative images are shown. **(D) Overexpressed PolγA colocalize with exogenously expressed MITOL.** U-2 OS cells were transfected with Flag PolγA WT and Myc MITOL. Cells were stained with anti-Flag and anti-Myc tag antibodies. Scale, 5 μM. Representative images are shown. **(E) Spacer and thumb domains of PolγA interacts with MITOL.** (Middle and bottom panels which constitute the input) Myc-MITOL expressed in HEK293T was detected by anti-Myc antibody and bound GST or GST-PolγA proteins [PolγA (53–439), PolγA (440–1239), PolγA (440–815), PolγA (816–1239), and PolγA (53–1239)] were detected by Coomassie staining. (Top) The interactions between Myc-MITOL and bound GST or GST-PolγA proteins were detected with anti-Myc tag antibody. Three independent biological replicates were carried out, and the same result was obtained. **(F) Carboxyl terminus loop of MITOL interacts with PolγA.** (Left panels, which constitute the input) PolγA (as visualized by western analysis of the $S^{35}$ methionine radiolabeled *in vitro* transcribed and translated product with anti-PolγA antibody) and bound GST or GST-MITOL [MITOL (1–278), MITOL (1–191), MITOL (159–210), and MITOL (253–278)] were visualized by Coomassie staining. (Right) Interaction was carried out between $S^{35}$ methionine radiolabeled PolγA and bound GST or GST-MITOL proteins. The amount of radiolabeled PolγA bound to the GST-tagged proteins was detected by autoradiography. Three independent biological replicates were carried out, and the same result was obtained. Numerical values for all graphs can be found in S1 Data. NHF, normal human fibroblast; PolγA, polymerase γ subunit A; RT-qPCR, reverse transcription quantitative polymerase chain reaction; WT, wild-type.
(PDF)

**S2 Fig. Factors which indicate the specificity of PolγA as a substrate for MITOL (related to Fig 2). (A) PolγA is not a substrate of HUWE1.** Lysates were made from HEK293T cells transfected with either siControl or siHUWE1 or siMITOL. Western blot analysis was carried out with the indicated antibodies. Three independent experiments were done, and the same results were obtained. **(B) Coomassie gel showing purified recombinant His GST USP30 and GST MITOL.** Coomassie gels indicating the purity of His GST USP30 and GST-MITOL used in the assays. Three independent protein preparations were used for the experiments. **(C, D) USP30 deubiquitylates PolγA *in vitro*.** *In vitro* ubiquitylation reactions were carried out using PolγA as the substrate and MITOL WT. Recombinant USP30 was added either (C) during the *in vitro* ubiquitylation assay (called simultaneous) or (D) after MITOL-mediated *in vitro* ubiquitylation assay (called sequential). Post-reaction, the products were detected by western blot analysis with the indicated antibodies. Three biological replicates were carried

out, and the same result was obtained. **(E) Overexpression of USP30 cannot revert MITOL-mediated degradation of PolγA.** Lysates were made from HEK293T cells transfected with either Flag USP30 or Myc MITOL WT. Western blot analysis was carried out with the indicated antibodies. Three independent experiments were done, and the same results were obtained. **(F) Overexpression of USP30 cannot revert MITOL-mediated ubiquitylation of PolγA.** Immunoprecipitations with either PolγA antibody (or the corresponding IgG) were carried out with lysates were made from HEK293T cells transfected with either Flag USP30 or Myc MITOL WT. Western blot analysis was carried out with the indicated antibodies. Three independent experiments were done, and the same results were obtained. IgG, immunoglobulin G; PolγA, polymerase γ subunit A; WT, wild-type.
(PDF)

**S3 Fig. Catalytically active MTOL can ubiquitylate PolγA via specific linkage (related to Fig 3). (A) Generation of ubiquitylated and non-ubiquitylated PolγA.** MITOL WT or CD dependent *in vitro* ubiquitylation reactions were carried out with PolγA WT or K1060R. The ubiquitylated products were detected by carrying out western blot analysis with anti-Ub (P4D1) antibodies. Equal amounts of substrates used in each condition was determined by carrying out westerns with anti-PolγA antibodies. Four independent biological replicates were carried out, and the same result was obtained. **(B) Time course of PolγA ubiquitylation by MITOL.** *In vitro* ubiquitylation reactions were carried out using PolγA as the substrate and MITOL WT as the E3 ligase. (Top) The ubiquitylation reactions were carried out for 5 minutes and 20 minutes. Post-reaction, the products were detected by western blot analysis with anti-PolγA antibody. (Bottom) Pre-reaction, the amount of PolγA protein used in each ubiquitylation reaction was determined by western analysis using anti-PolγA antibodies. Three independent biological replicates were carried out, and the same result was obtained. **(C) PolγA require catalytically active MITOL during *in vitro* ubiquitylation.** *In vitro* ubiquitylation reactions were carried out using $S^{35}$ methionine radiolabeled PolγA as the substrate and MITOL WT or CD as the E3 ligase. Post-ubiquitylation, the products were detected by carrying out western blot analysis with anti-PolγA antibodies. The reaction products where PolγA were ubiquitylated by MITOL WT were designated as Ub PolγA. Alternately, the products obtained when MITOL CD was used were designated as Non Ub PolγA. The amount of PolγA used in each condition was determined by western blotting with antibodies against PolγA. Three independent biological replicates were carried out, and the same result was obtained. **(D) Purity of GST-Tom20.** Coomassie gels indicating the purity of GST and GST-Tom20 used in the assays. Three independent protein preparations were used for the experiments. **(E) PolγA was ubiquitylated by Ub K6O and not by Ub K6R.** Same as (C) except MITOL-dependent ubiquitylation, reactions for PolγA were carried out with 2 ubiquitin variants—Ub K6R or Ub K6O. Three independent biological replicates were carried out, and the same result was obtained. CD, catalytically dead; PolγA, polymerase γ subunit A; WT, wild-type.
(PDF)

**S4 Fig. Controls showing specificity of the entry of a subset of PEO mutants into mitochondria (related to Fig 4). (A) Levels of PolγA WT and PEO mutants.** *In vitro* transcribed and translated PolγA WT, PEO mutant #1, PEO mutant #2, PEO mutant #3, and PEO mutant #4 were subjected to SDS-PAGE, and the products were detected by autoradiography. Three independent biological replicates were carried out, and the same result was obtained. **(B) PEO patient #4 shows similar extent of mitochondrial import as PolγA WT.** Mitochondrial import assay was carried out using the indicated mitochondrial fractions. $S^{35}$ methionine radiolabeled PolγA WT, PEO mutant #4 were incubated with each of the mitochondrial fractions. (Bottom) The purity of the mitochondrial fractions was determined using the indicated

antibodies. Three independent biological replicates were carried out, and the same result was obtained. **(C) Relative levels of endogenous and exogenous PolγA WT and PEO mutants.** Lysates were made from HEK293T cells transfected with Flag PolγA WT, Flag PEO mutant #1, Flag PEO mutant #2, Flag PEO mutant #3, and Flag PEO mutant #4. Western blot analysis was carried out with the indicated antibodies. The relative levels of PolγA to hsp60 have been quantitated from 3 biological replicates. **(D) PEO patient #4 shows similar extent of ubiquitylation and binding to Tom20 as PolγA WT.** (Left) Whole cell extracts were made from HEK293T cells transfected with His-Ub, Myc MITOL WT, PolγA WT, and PEO mutant #4. Western blot analysis was carried out with the indicated antibodies. (Right) Immunoprecipitations were carried out with anti-Flag antibody (or the corresponding IgG), and the immunoprecipitates were probed with the indicated antibodies. Three independent biological replicates were carried out, and the same result was obtained. Numerical values for all graphs can be found in S1 Data. His-Ub, His-tagged ubiquitin; IgG, immunoglobulin G; PEO, progressive external ophthalmoplegia; PolγA, polymerase γ subunit A; WT, wild-type. (PDF)

**S5 Fig. Effect of PEO mutants during PolγA-mediated in vitro polymerase and in vitro exonuclease activities (related to Fig 5).** (A) Purity of proteins used in primer extension and exonuclease assays. Coomassie gels indicating the purity of GST-tagged PolγA WT, PEO mutant #1, PEO mutant #2, PEO mutant #3, PEO mutant #4, and GST-PolγB used in the primer extension and exonuclease assays. Two independent protein preparations were used for the experiments. (B, C) All PEO mutants show compromised *in vitro* polymerase activity. (B) *In vitro* primer extension assays were carried out with PolγA WT, PEO mutant #1, PEO mutant #2, PEO mutant #3, and PEO mutant #4 using the primer-template pair in presence of $[\alpha\text{-}P^{32}]$ dATP. (C) The 40-bp radiolabeled product was quantitated from 3 biological replicates. (D, E) Except PEO mutant #4, all other PEO mutants show compromised *in vitro* exonuclease activity. (D) *In vitro* exonuclease assays were carried out with PolγA WT, PEO mutant #1, PEO mutant #2, PEO mutant #3, and PEO mutant #4 using the $[\gamma\text{-}^{32}P]$ ATP labeled primer annealed to the template. (E) The smaller products obtained were quantitated from 3 biological replicates. Numerical values for all graphs can be found in S1 Data. PEO, progressive external ophthalmoplegia; PolγA, polymerase γ subunit A; WT, wild-type. (PDF)

**S6 Fig. Factors discriminating the entry of PEO mutants into mitochondria (related to Figs 4 and 6). (A, B) Cells expressing PEO mutant #1 and #2 have lower mtDNA integrity.** (A) mtDNA integrity was measured using a long-range mtDNA amplification assay in HEK293T cells expressing Flag PolγA WT, Flag PEO mutant #1, and Flag PEO mutant #2. Amplification of mtND1 was used as a control. (B) Quantitation of (A), done with data from 3 biological replicates. **(C, D) Subsets of PEO mutants are present in the insoluble fraction.** (C) Soluble and insoluble fractions were made from HEK293T cells overexpressing Flag PolγA WT, Flag PEO mutant #1, Flag PEO mutant #2, Flag PEO mutant #3, and Flag PEO mutant #4. Western blot analysis was carried out with the indicated antibodies. Equal amounts of the soluble and insoluble fractions were visualized by Coomassie staining. (D) Quantitation of (C), done with data from 3 biological replicates. **(E, F) K1060R mutation in PEO mutants #1 and #2 leads to their decreased presence in the insoluble fraction.** (E) Soluble and insoluble fractions were made from HEK293T cells overexpressing Flag PEO mutant #1, Flag PEO mutant #1 K1060R, Flag PEO mutant #2, and Flag PEO mutant #2 K1060R. Western blot analysis was carried out with the indicated antibodies. (F) Quantitation of (E), done with data from 3 biological replicates. Numerical values for all graphs can be found in S1 Data. mtDNA, mitochondrial DNA; PEO, progressive external ophthalmoplegia; PolγA, polymerase γ subunit

A; WT, wild-type.
(PDF)

**S1 Table. Details of patient mutations used in the study.**
(PDF)

**S2 Table. List of antibodies used in the study.**
(PDF)

**S3 Table. List of recombinant DNAs used in the study.**
(PDF)

**S4 Table. List of reagents used in the study.**
(PDF)

**S5 Table. List of primers used in the study.**
(PDF)

**S6 Table. Statistical analysis performed in this study.**
(PDF)

**S1 Data. Numerical raw data. All numerical raw data are combined in a single PDF file, "S1_Data."** The presented data underlays Figs 1ACDEFGI, 3BDFHI, 4BEG, 5AD, and 6CE and S1AB, S4C, S5CE, and S6BDF Figs.
(PDF)

**S1 Original Images. Original gel and images contained in this manuscript, related to Figs 1ABCDEFGHJ, 2ABCDEFG, 3ACEGHI, 4ACDEF, 5ABC, and 6ABD and S1ADEF, S2ABCDEF, S3ABCDE, S4ABCD, S5ABD, and S6ACE Figs.**
(PDF)

## Acknowledgments

The authors acknowledge Shigeru Yanagi (Tokyo University of Pharmacy and Life Sciences, Japan), Quan Chen (State Key Laboratory of Membrane Biology, China), Akhil Banerjea, (National Institute of Immunology, India), and Daisuke Kohda (Kyushu University, Japan) for recombinants and Shigeru Yanagi (Tokyo University of Pharmacy and Life Sciences, Japan) for cells and MITOL antibody.

## Author Contributions

**Conceptualization:** Sagar Sengupta.

**Data curation:** Mansoor Hussain, Aftab Mohammed, Shabnam Saifi, Aamir Khan, Ekjot Kaur, Swati Priya, Himanshi Agarwal.

**Formal analysis:** Mansoor Hussain, Sagar Sengupta.

**Funding acquisition:** Sagar Sengupta.

**Investigation:** Mansoor Hussain, Aftab Mohammed, Shabnam Saifi, Aamir Khan, Ekjot Kaur, Swati Priya, Himanshi Agarwal.

**Methodology:** Mansoor Hussain, Aftab Mohammed, Shabnam Saifi, Aamir Khan, Ekjot Kaur, Swati Priya, Himanshi Agarwal.

**Project administration:** Sagar Sengupta.

**Resources:** Sagar Sengupta.

**Supervision:** Sagar Sengupta.

**Validation:** Mansoor Hussain, Aftab Mohammed, Shabnam Saifi, Aamir Khan, Ekjot Kaur, Swati Priya, Himanshi Agarwal.

**Visualization:** Sagar Sengupta.

**Writing – original draft:** Sagar Sengupta.

**Writing – review & editing:** Sagar Sengupta.

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
