## [Editor Report · Decision Letter 0]

3 Sep 2020

Dear Dr Sengupta, 

Thank you for submitting your manuscript entitled "MITOL dependent ubiquitylation negatively regulates the entry of PolGA into mitochondria" for consideration as a Research Article by PLOS Biology.

Your manuscript has now been evaluated by the PLOS Biology editorial staff [as well as by an academic editor with relevant expertise] and I am writing to let you know that we would like to send your submission out for external peer review.

Please re-submit your manuscript within two working days, i.e. by Sep 05 2020 11:59PM.

Kind regards,

Maya Capelson,

PLOS Biology

---

## [Editor Report · Decision Letter 1]

14 Sep 2020

Dear Dr Sengupta,

Thank you very much for submitting your manuscript "MITOL dependent ubiquitylation negatively regulates the entry of PolGA into mitochondria" for consideration as a Research Article at PLOS Biology. As you know, your manuscript and plan of revision have been evaluated by the PLOS Biology editors and by an Academic Editor with relevant expertise.

Based on your responses to the reviews from Reviews Commons, we would welcome re-submission of a revised version that takes into account the reviewers' comments. We cannot make any decision about publication until we have seen the revised manuscript and your response to the reviewers' comments. Your revised manuscript is also likely to be sent for further evaluation by the original reviewers.

We expect to receive your revised manuscript within 3 months. 

**IMPORTANT - SUBMITTING YOUR REVISION**

*Re-submission Checklist*

*Published Peer Review*

*PLOS Data Policy*

*Blot and Gel Data Policy*

Sincerely,

Ines

--

Ines Alvarez-Garcia, PhD,

Senior Editor,

ialvarez-garcia@plos.org,

PLOS Biology

---

## [Decision Letter · Decision Letter 2]

22 Jan 2021

Dear Dr Sengupta,

Thank you for submitting your revised Research Article entitled "MITOL dependent ubiquitylation negatively regulates the entry of PolgA into mitochondria" for publication in PLOS Biology. I have now obtained advice from the original reviewers and have discussed their comments with the Academic Editor. 

Based on the reviews, we will probably accept this manuscript for publication, assuming that you will modify the manuscript to address the data and other policy-related requests noted at the end of this email. We would also suggest to add a hyphen to the title as follows:

"MITOL-dependent ubiquitylation negatively regulates the entry of PolgA into mitochondria"

We expect to receive your revised manuscript within two weeks.

-  a cover letter that should detail your responses to any editorial requests.

*Published Peer Review History*

*Early Version*

Sincerely,

Ines

--

Ines Alvarez-Garcia, PhD,

Senior Editor,

PLOS Biology

Fig. 1A, C-G, I; Fig. 3B, D, F, H, I; Fig. 4C, E, G; Fig. 5A, D; Fig. 6C, E; Fig. S1A, B; Fig. S4C; Fig. S5C, E and Fig. S6B, D, F

BLURB

Please also provide a blurb which (if accepted) will be included in our weekly and monthly Electronic Table of Contents, sent out to readers of PLOS Biology, and may be used to promote your article in social media. The blurb should be about 30-40 words long and is subject to editorial changes. It should, without exaggeration, entice people to read your manuscript. It should not be redundant with the title and should not contain acronyms or abbreviations. For examples, view our author guidelines: https://journals.plos.org/plosbiology/s/revising-your-manuscript#loc-blurb

Reviewers’ comments

Rev. 1:

The authors have made excellent efforts resolving the raised issues. I have no further comments to add.

Rev. 2:

The authors adequately addressed the major comments raised by the reviewers. The additional data on functional effects of MITOL knockdown on mitochondrial import further strengthened the manuscript. The manuscript should be in good shape for publication in Plos Biology.

Rev. 3:

I am satisfied with the author's response to my comments.

---

## [Editor Report · Decision Letter 3]

3 Feb 2021

Dear Dr Sengupta,

Thank you for submitting your revised Research Article entitled "MITOL-dependent ubiquitylation negatively regulates the entry of PolgA into mitochondria" for publication in PLOS Biology.

Thank you for addressing the pending editorial requests. We are almost satisfied with the manuscript, but we would like you to indicate in all the corresponding figure legends (including the supplementary figures) where the underlying data can be found.

Please resubmit the manuscript within 1 week.

Please don't hesitate to contact me if you have any questions.

Sincerely,

Ines

--

Ines Alvarez-Garcia, PhD,

Senior Editor,

PLOS Biology

---

## [Editor Report · Decision Letter 4]

4 Feb 2021

Dear Dr Sengupta,

On behalf of my colleagues and the Academic Editor, Rong Tian, I am pleased to say that we can in principle offer to publish your Research Article entitled "MITOL-dependent ubiquitylation negatively regulates the entry of PolgA into mitochondria" in PLOS Biology, provided you address any remaining formatting and reporting issues. These will be detailed in an email that will follow this letter and that you will usually receive within 2-3 business days, during which time no action is required from you. Please note that we will not be able to formally accept your manuscript and schedule it for publication until you have made the required changes.

PRESS

Thank you again for supporting Open Access publishing. We look forward to publishing your paper in PLOS Biology. 

Sincerely, 

Ines

--

Ines Alvarez-Garcia, PhD 

Senior Editor 

PLOS Biology
